# Kernelized Wasserstein Natural Gradient

**Michael Arbel, Arthur Gretton**
Gatsby Computational Neuroscience Unit
University College London
`{michael.n.arbel,arthur.gretton}@gmail.com`

**Wuchen Li**
University of California, Los Angeles
`wcli@math.ucla.edu`

**Guido Montúfar**
University of California, Los Angeles, and Max Planck Institute for Mathematics in the Sciences
`montufar@mis.mpg.de`

## Abstract

Many machine learning problems can be expressed as the optimization of some cost functional over a parametric family of probability distributions. It is often beneficial to solve such optimization problems using natural gradient methods. These methods are invariant to the parametrization of the family, and thus can yield more effective optimization. Unfortunately, computing the natural gradient is challenging as it requires inverting a high dimensional matrix at each iteration. We propose a general framework to approximate the natural gradient for the Wasserstein metric, by leveraging a dual formulation of the metric restricted to a Reproducing Kernel Hilbert Space. Our approach leads to an estimator for gradient direction that can trade-off accuracy and computational cost, with theoretical guarantees. We verify its accuracy on simple examples, and show the advantage of using such an estimator in classification tasks on `Cifar10` and `Cifar100` empirically.

## 1 Introduction

The success of machine learning algorithms relies on the quality of an underlying optimization method. Many of the current state-of-the-art methods rely on variants of Stochastic Gradient Descent (SGD) such as AdaGrad (Duchi et al., 2011), RMSProp (Hinton et al., 2012), and Adam (Kingma and Ba, 2014). While generally effective, the performance of such methods remains sensitive to the curvature of the optimization objective. When the Hessian matrix of the objective at the optimum has a large condition number, the problem is said to have a pathological curvature (Martens, 2010; Sutskever et al., 2013). In this case, the first-order optimization methods tend to have poor performance. Using adaptive step sizes can help when the principal directions of curvature are aligned with the coordinates of the vector parameters. Otherwise, an additional rotation of the basis is needed to achieve this alignment. One strategy is to find an alternative parametrization of the same model that has a better-behaved curvature and is thus easier to optimize with standard first-order optimization methods. Designing good network architectures (Simonyan and Zisserman, 2014; He et al., 2015) along with normalization techniques (LeCun et al., 2012; Ioffe and Szegedy, 2015; Salimans and Kingma, 2016) is often critical for the success of such optimization methods.

The natural gradient method (Amari, 1998) takes a related but different perspective. Rather than re-parametrizing the model, the natural gradient method tries to make the optimizer itself invariant to re-parameterizations by directly operating on the manifold of probability distributions. This requires endowing the parameter space with a suitable notion of proximity formalized by a metric. An important metric in this context is the Fisher information metric (Fisher and Russell, 1922; Rao, 1992), which induces the Fisher-Rao natural gradient (Amari, 1985). Another important metric in probability space is the Wasserstein metric (Villani, 2009; Otto, 2001), which induces the Wasserstein natural gradient (Li and Montufar, 2018a;b; Li, 2018); see similar formulations in Gaussian families (Malagò et al., 2018; Modin, 2017). In spite of their numerous theoretical advantages, applying natural gradient methods is challenging in practice. Indeed, each parameter update requires inverting the metric tensor. This becomes infeasible for current deep learning models, which typically have millions of parameters. This has motivated research into finding efficient algorithms to estimate the natural gradient (Martens and Grosse, 2015; Grosse and Martens, 2016; George et al., 2018;

Heskes, 2000; Bernacchia et al., 2018). Such algorithms often address the case of the Fisher metric and either exploit a particular structure of the parametric family or rely on a low rank decomposition of the information matrix. Recently, Li et al. (2019) proposed to estimate the metric based on a dual formulation and used this estimate in a proximal method. While this avoids explicitly computing the natural gradient, the proximal method also introduces an additional optimization problem to be solved at each update of the model's parameters. The quality of the solver will thus depend on the accuracy of this additional optimization.

In this paper, we use the dual formulation of the metric to directly obtain a closed form expression of the natural gradient as a solution to a convex functional optimization problem. We focus on the Wasserstein metric as it has the advantage of being well defined even when the model doesn't admit a density. The expression remains valid for general metrics including the Fisher-Rao metric. We leverage recent work on Kernel methods (Sriperumbudur et al., 2017; Arbel and Gretton, 2017; Sutherland et al., 2017; Mroueh et al., 2019) to compute an estimate of the natural gradient by restricting the functional space appearing in the dual formulation to a Reproducing Kernel Hilbert Space. We demonstrate empirically the accuracy of our estimator on toy examples, and show how it can be effectively used to approximate the trajectory of the natural gradient descent algorithm. We also analyze the effect of the dimensionality of the model on the accuracy of the proposed estimator. Finally, we illustrate the benefits of our proposed estimator for solving classification problems when the model has an ill-conditioned parametrization.

The paper is organized as follows. In Section 2, after a brief description of natural gradients, we discuss Legendre duality of metrics, and provide details on the Wasserstein natural gradient. In Section 3, we present our kernel estimator of the natural gradient. In Section 4 we present experiments to evaluate the accuracy of the proposed estimator and demonstrate its effectiveness in supervised learning tasks.

## 2    NATURAL GRADIENT DESCENT

We first briefly recall the natural gradient descent method in Section 2.1, and its relation to metrics on probability distribution spaces in Section 2.2. We next present Legendre dual formulations for metrics in Section 2.3 where we highlight the Fisher-Rao and Wasserstein metrics as important examples.

### 2.1    GENERAL FORMULATION

It is often possible to formulate learning problems as the minimization of some cost functional $\rho \mapsto \mathcal{F}(\rho)$ over probability distributions $\rho$ from a parametric model $\mathcal{P}_\Theta$. The set $\mathcal{P}_\Theta$ contains probability distributions defined on an open sample space $\Omega \subset \mathbb{R}^d$ and parametrized by some vector $\theta \in \Theta$, where $\Theta$ is an open subset of $\mathbb{R}^q$. The learning problem can thus be formalized as finding an optimal value $\theta^*$ that locally minimizes a loss function $\mathcal{L}(\theta) := \mathcal{F}(\rho_\theta)$ defined over the parameter space $\Theta$. One convenient way to solve this problem approximately is by gradient descent, which uses the *Euclidean gradient* of $\mathcal{L}$ w.r.t. the parameter vector $\theta$ to produce a sequence of updates $\theta_t$ according to the following rule:

$$\theta_{t+1} = \theta_t - \gamma_t \nabla \mathcal{L}(\theta_t).$$

Here the step-size $\gamma_t$ is a positive real number. The *Euclidean gradient* can be viewed as the direction in parameter space that leads to the highest decrease of some *linear model* $\mathcal{M}_t$ of the cost function $\mathcal{L}$ per unit of change of the parameter. More precisely, the *Euclidean gradient* is obtained as the solution of the optimization problem:

$$\nabla \mathcal{L}(\theta_t) = -\underset{u \in \mathbb{R}^q}{\operatorname{argmin}} \; \mathcal{M}_t(u) + \frac{1}{2}\|u\|^2. \tag{1}$$

The linear model $\mathcal{M}_t$ is an approximation of the cost function $\mathcal{L}$ in the neighborhood of $\theta_t$ and is simply obtained by a first order expansion: $\mathcal{M}_t(u) = \mathcal{L}(\theta_t) + \nabla \mathcal{L}(\theta_t)^\top u$. The quadratic term $\|u\|^2$ penalizes the change in the parameter and ensures that the solution remains in the neighborhood where the linear model is still a good approximation of the cost function.

This particular choice of quadratic term is what defines the *Euclidean gradient* descent algorithm, which can often be efficiently implemented for neural network models using *back-propagation*. The performance of this algorithm is highly dependent on the parametrization of the model $\mathcal{P}_\Theta$, however (Martens, 2010; Sutskever et al., 2013). To obtain an algorithm that is robust to parametrization, one can take advantage of the structure of the cost function $\mathcal{L}(\theta)$ which is obtained as the composition of the functional $\mathcal{F}$ and the model $\theta \mapsto \rho_\theta$ and define a *generalized natural gradient* (Amari and Cichocki, 2010). We first provide

a conceptual description of the general approach to obtain such gradient. The starting point is to choose a divergence $D$ between probability distributions and use it as a new penalization term:

$$-\operatorname*{argmin}_{u\in\mathbb{R}^q} \mathcal{M}_t(u) + \frac{1}{2} D(\rho_{\theta_t}, \rho_{\theta_t+u}). \tag{2}$$

Here, changes in the model are penalized directly in probability space rather than parameter space as in (1). In the limit of small $u$, the penalization term can be replaced by a quadratic term $u^\top G_D(\theta)u$ where $G_D(\theta)$ contains second order information about the model as measured by $D$. This leads to the following expression for the *generalized natural gradient* $\nabla^D \mathcal{L}(\theta_t)$ where the dependence in $D$ is made explicit:

$$\nabla^D \mathcal{L}(\theta_t) := -\operatorname*{argmin}_{u\in\mathbb{R}^q} \mathcal{M}_t(u) + \frac{1}{2} u^\top G_D(\theta_t)u. \tag{3}$$

From (3), it is possible to express the *generalized natural gradient* by means of the *Euclidean gradient*: $\nabla^D \mathcal{L}(\theta_t) = G_D(\theta_t)^{-1} \nabla \mathcal{L}(\theta_t)$. The parameter updates are then obtained by the new update rule:

$$\theta_{t+1} = \theta_t - \gamma_t G_D(\theta_t)^{-1} \nabla \mathcal{L}(\theta_t). \tag{4}$$

Equation (4) leads to a descent algorithm which is invariant to parametrization in the continuous-time limit:

**Proposition 1.** *Let $\Psi$ be an invertible and smoothly differentiable re-parametrization $\psi = \Psi(\theta)$ and denote by $\bar{\mathcal{L}}(\psi) := \mathcal{L}(\Psi^{-1}(\psi))$. Consider the continuous-time natural gradient flows:*

$$\dot{\theta}_s = -\nabla^D_\theta \mathcal{L}(\theta_s), \qquad \dot{\psi}_s = -\nabla^D_\psi \bar{\mathcal{L}}(\psi_s), \qquad \psi_0 = \Psi(\theta_0)$$

*Then $\psi_s$ and $\theta_s$ are related by the equation $\psi_s = \Psi(\theta_s)$ at all times $s \geq 0$.*

This result implies that an ill-conditioned parametrization of the model has little effect on the optimization when (4) is used. It is a consequence of the transformation properties of the natural gradient by change of parametrization: $\nabla^D_\psi \bar{\mathcal{L}}(\psi) = \nabla_\theta \Psi(\theta) \nabla^D_\theta \mathcal{L}(\theta)$ which holds in general for any covariant gradient. We provide a proof of Proposition 1 in Appendix C.1 in the particular the case when $D$ is either Kullback-Leibler divergence $F$, or the squared Wasserstein-2 distance $W$ using notions introduced later in Section 2.3 and refer to Ollivier et al. (2011) for a detailed discussion.

The approach based on (2) for defining the generalized natural gradient is purely conceptual and can be formalized using the notion of metric tensor from differential geometry which allows for more generality. In Section 2.2, we provide such formal definition in the case when $D$ is either the Kullback-Leibler divergence $F$, or the squared Wasserstein-2 distance $W$.

## 2.2 INFORMATION MATRIX VIA DIFFERENTIAL GEOMETRY

When $D$ is the Kullback-Leibler divergence or relative entropy $F$, then (3) defines the *Fisher-Rao natural gradient* $\nabla^F \mathcal{L}(\theta)$ (Amari, 1985) and $G_F(\theta)$ is called the *Fisher information matrix*. $G_F(\theta)$ is well defined when the probability distributions in $\mathcal{P}_\Theta$ all have positive densities, and when some additional differentiability and integrability assumptions on $\rho_\theta$ are satisfied. In fact, it has an interpretation in Riemannian geometry as the pull-back of a metric tensor $g^F$ defined over the set of probability distributions with positive densities and known as the *Fisher-Rao metric* (see Definition 4 in Appendix B.1; see also Holbrook et al. 2017):

**Definition 1** (Fisher information matrix). *Assume $\theta \mapsto \rho_\theta(x)$ is differentiable for all $x$ on $\Omega$ and that $\int \frac{\|\nabla \rho_\theta(x)\|^2}{\rho_\theta(x)} \mathrm{d}x < \infty$. Then the Fisher information matrix is defined as the pull-back of the Fisher-Rao metric $g^F$:*

$$G_F(\theta)_{ij} = g^F_{\rho_\theta}(\partial_i \rho_\theta, \partial_j \rho_\theta) := \int f_i(x) f_j(x) \rho_\theta(x) dx,$$

*where the functions $f_i$ on $\Omega$ are given by: $f_i = \frac{\partial_i \rho_\theta}{\rho_\theta}$.*

Definition 1 directly introduces $G_F$ using the *Fisher-Rao metric* tensor which captures the infinitesimal behavior of the KL. This approach can be extended to any metric tensor $g$ defined on a suitable space of probability distributions containing $\mathcal{P}_\Theta$. In particular, when $D$ is the Wasserstein-2, the *Wasserstein information matrix* is obtained directly by means of the Wasserstein-2 metric tensor $g^W$ (Otto and Villani, 2000; Lafferty and Wasserman, 2008) as proposed in Li and Montufar (2018a); Chen and Li (2018):

**Definition 2** (Wasserstein information matrix). *The Wasserstein information matrix (WIM) is defined as the pull-back of the Wasserstein 2 metric $g^W$ :*

$$G_W(\theta)_{ij} = g^W_{\rho_\theta}(\partial_i \rho_\theta, \partial_j \rho_\theta) := \int \phi_i(x)^\top \phi_j(x) \mathrm{d}\rho_\theta(x),$$

*where $\phi_i$ are vector valued functions on $\Omega$ that are solutions to the partial differential equations with Neumann boundary condition:*

$$\partial_i \rho_\theta = -div(\rho_\theta \phi_i), \qquad \forall 1 \le i \le q.$$

*Moreover, $\phi_i$ are required to be in the closure of the set of gradients of smooth and compactly supported functions in $L_2(\rho_\theta)^d$. In particular, when $\rho_\theta$ has a density, $\phi_i = \nabla_x f_i$, for some real valued function $f_i$ on $\Omega$.*

The partial derivatives $\partial_i \rho_\theta$ should be understood in distribution sense, as discussed in more detail in Section 2.3. This allows to define the *Wasserstein natural gradient* even when the model $\rho_\theta$ does not admit a density. Moreover, it allows for more generality than the conceptual approach based on (2) which would require performing a first order expansion of the Wasserstein distance in terms of its *linearized version* known as the *Negative Sobolev distance*. We provide more discussion of those two approaches and their differences in Appendix B.3. From now on, we will focus on the above two cases of the natural gradient $\nabla^D \mathcal{L}(\theta)$, namely $\nabla^F \mathcal{L}(\theta)$ and $\nabla^W \mathcal{L}(\theta)$. When the dimension of the parameter space is high, directly using equation (4) becomes impractical as it requires storing and inverting the matrix $G(\theta)$. In Section 2.3 we will see how equation (3) can be exploited along with Legendre duality to get an expression for the natural gradient that can be efficiently approximated using kernel methods.

## 2.3 LEGENDRE DUALITY FOR METRICS

In this section we provide an expression for the *natural gradient* defined in (3) as the solution of a saddle-point optimization problem. It exploits Legendre duality for metrics to express the quadratic term $u^\top G(\theta)u$ as a solution to a functional optimization problem over $C_c^\infty(\Omega)$, the set of smooth and compactly supported functions on $\Omega$. The starting point is to extend the notion of gradient $\nabla \rho_\theta$ which appears in Definitions 1 and 2 to the distributional sense of Definition 3 below.

**Definition 3.** *Given a parametric family $\mathcal{P}_\Theta$ of probability distributions, we say that $\rho_\theta$ admits a distributional gradient at point $\theta$ if there exists a linear continuous map $\nabla \rho_\theta : C_c^\infty(\Omega) \to \mathbb{R}^q$ such that:*

$$\int f(x) \mathrm{d}\rho_{\theta+\epsilon u}(x) - \int f(x) \mathrm{d}\rho_\theta(x) = \epsilon \nabla \rho_\theta(f)^\top u + \epsilon \delta(\epsilon, f, u) \qquad \forall f \in C_c^\infty(\Omega), \quad \forall u \in \mathbb{R}^q$$

*where $\delta(\epsilon, f, u)$ depends on $f$ and $u$ and converges to 0 as $\epsilon$ approaches 0. $\nabla \rho_\theta$ is called the distributional gradient of $\rho_\theta$ at point $\theta$.*

When the distributions in $\mathcal{P}_\Theta$ have a density, written $x \mapsto \rho_\theta(x)$ by abuse of notation, that is differentiable w.r.t. $\theta$ and with a jointly continuous gradient in $\theta$ and $x$ then $\nabla \rho_\theta(f)$ is simply given by $\int f(x) \nabla_\theta \rho_\theta(x) \mathrm{d}x$ as shown in Proposition 12 of Appendix C.1. In this case, the *Fisher-Rao natural gradient* admits a formulation as a saddle point solution involving $\nabla \rho_\theta$ and provided in Proposition 2 with a proof in Appendix C.1.

**Proposition 2.** *Under the same assumptions as in Definition 1, the Fisher information matrix admits the dual formulation:*

$$\frac{1}{2} u^\top G_F(\theta) u := \sup_{\substack{f \in C_c^\infty(\Omega) \\ \int f(x) \mathrm{d}\rho_\theta(x) = 0}} \nabla \rho_\theta(f)^\top u - \frac{1}{2} \int f(x)^2 \mathrm{d}\rho_\theta(x) \mathrm{d}x. \tag{5}$$

*Moreover, defining $\mathcal{U}_\theta(f) := \nabla \mathcal{L}(\theta) + \nabla \rho_\theta(f)$, the Fisher-Rao natural gradient satisfies:*

$$\nabla^F \mathcal{L}(\theta) = -\operatorname*{argmin}_{u \in \mathbb{R}^q} \sup_{\substack{f \in C_c^\infty(\Omega) \\ \int f(x) \mathrm{d}\rho_\theta(x) = 0}} \mathcal{U}_\theta(f)^\top u - \frac{1}{2} \int f(x)^2 \mathrm{d}\rho_\theta(x) \mathrm{d}x,$$

Another important case is when $\mathcal{P}_\Theta$ is defined as an implicit model. In this case, any sample $x$ from a distribution $\rho_\theta$ in $\mathcal{P}_\Theta$ is obtained as $x = h_\theta(z)$, where $z$ is a sample from a fixed latent distribution $\nu$

defined over a latent space $\mathcal{Z}$ and $(\theta,z) \mapsto h_\theta(z)$ is a deterministic function with values in $\Omega$. This can be written in a more compact way as the push-forward of $\nu$ by the function $h_\theta$:

$$\mathcal{P}_\Theta := \{\rho_\theta := (h_\theta)_\# \nu \,|\, \theta \in \Omega\}. \tag{6}$$

A different expression for $\nabla \rho_\theta$ is obtained in the case of implicit models when $\theta \mapsto h_\theta(z)$ is differentiable for $\nu$-almost all $z$ and $\nabla h_\theta$ is square integrable under $\nu$:

$$\nabla \rho_\theta(f) = \int \nabla h_\theta(z)^\top \nabla_x f(h_\theta(z)) \mathrm{d}\nu(z). \tag{7}$$

Equation (7) is also known as the re-parametrization trick (Kingma et al., 2015) and allows to derive a dual formulation of the *Wasserstein natural gradient* in the case of implicit models. Proposition 3 below provides such formulation under mild assumptions stated in Appendix A.2 along with a proof in Appendix C.1.

**Proposition 3.** *Assume $\mathcal{P}_\Theta$ is defined by (6) such that $\nabla \rho_\theta$ is given by (7). Under Assumptions (**B**) and (**C**), the Wasserstein information matrix satisfies:*

$$\frac{1}{2} u^\top G_W(\theta) u = \sup_{f \in C_c^\infty(\Omega)} \nabla \rho_\theta(f)^\top u - \frac{1}{2} \int \|\nabla_x f(x)\|^2 \mathrm{d}\rho_\theta(x) \tag{8}$$

*and the Wasserstein natural gradient satisfies:*

$$\nabla^W \mathcal{L}(\theta) = -\operatorname*{argmin}_{u \in \mathbb{R}^q} \sup_{f \in C_c^\infty(\Omega)} \mathcal{U}_\theta(f)^\top u - \frac{1}{2} \int \|\nabla_x f(x)\|^2 \mathrm{d}\rho_\theta(x). \tag{9}$$

The similarity between the variational formulations provided in Propositions 2 and 3 is worth noting. A first difference however, is that Proposition 3 doesn't require the test functions $f$ to have $0$ mean under $\rho_\theta$. This is due to the form of the objective in (8) which only depends on the gradient of $f$. More importantly, while (8) is well defined, the expression in (5) can be infinite when $\nabla \rho_\theta$ is given by (7). Indeed, if the $\rho_\theta$ doesn't admit a density, it is always possible to find an admissible function $f \in C_c^\infty(\Omega)$ with bounded second moment under $\rho_\theta$ but for which $\nabla \rho_\theta(f)$ is arbitrarily large. This is avoided in (8) since the quadratic term directly penalizes the gradient of functions instead. For similar reasons, the dual formulation of the Sobolev distance considered in Mroueh et al. (2019) can also be infinite in the case of implicit models as discussed in Appendix B.3 although formally similar to (8). Nevertheless, a similar estimator as in Mroueh et al. (2019) can be considered using kernel methods which is the object of Section 3.

## 3 KERNELIZED WASSERSTEIN NATURAL GRADIENT

In this section we propose an estimator for the Wasserstein natural gradient using kernel methods and exploiting the formulation in (9). We restrict to the case of the Wasserstein natural gradient (WNG), denoted by $\nabla^W \mathcal{L}(\theta)$, as it is well defined for implicit models, but a similar approach can be used for the Fisher-Rao natural gradient in the case of models with densities. We first start by presenting the *kernelized Wasserstein natural gradient* (KWNG) in Section 3.1, then we introduce an efficient estimator for KWNG in Section 3.2. In Section 3.4 we provide statistical guarantees and discuss practical considerations in Section 3.3.

### 3.1 GENERAL FORMULATION AND MINIMAX THEOREM

Consider a Reproducing Kernel Hilbert Space (RKHS) $\mathcal{H}$ which is a Hilbert space endowed with an inner product $\langle .,.\rangle_\mathcal{H}$ along with its norm $\|.\|_\mathcal{H}$. $\mathcal{H}$ has the additional property that there exists a symmetric positive semi-definite kernel $k : \Omega \times \Omega \mapsto \mathbb{R}$ such that $k(x,.) \in \mathcal{H}$ for all $x \in \Omega$ and satisfying the *Reproducing property* for all functions $f$ in $\mathcal{H}$:

$$f(x) = \langle f, k(x,.)\rangle_\mathcal{H}, \qquad \forall x \in \Omega. \tag{10}$$

The above property is central in all kernel methods as it allows to obtain closed form expressions for some class of functional optimization problems. In order to take advantage of such property for estimating the natural gradient, we consider a new saddle problem obtained by restricting (9) to functions in the RKHS $\mathcal{H}$ and adding some regularization terms:

$$\widetilde{\nabla}^W \mathcal{L}(\theta) := -\min_{u \in \mathbb{R}^q} \sup_{f \in \mathcal{H}} \mathcal{U}_\theta(f)^\top u - \frac{1}{2} \int \|\nabla_x f(x)\|^2 \mathrm{d}\rho_\theta(x) + \frac{1}{2}(\epsilon u^\top D(\theta) u - \lambda \|f\|_\mathcal{H}^2). \tag{11}$$

The *kernelized Wasserstein natural gradient* is obtained by solving (11) and is denoted by $\widetilde{\nabla}^W \mathcal{L}(\theta)$. Here, $\epsilon$ is a positive real numbers, $\lambda$ is non-negative while $D(\theta)$ is a diagonal matrix in $\mathbb{R}^q$ with positive diagonal elements whose choice will be discussed in Section 3.3. The first regularization term makes the problem strongly convex in $u$, while the second term makes the problem strongly concave in $f$ when $\lambda > 0$. When $\lambda = 0$, the problem is still concave in $f$. This allows to use a version of the minimax theorem (Ekeland and Témam, 1999, Proposition 2.3, Chapter VI) to exchange the order of the supremum and minimum which also holds true when $\lambda = 0$. A new expression for the kernelized natural gradient is therefore obtained:

**Proposition 4.** *Assume that $\epsilon > 0$ and $\lambda > 0$, then the kernelized natural gradient is given by:*

$$\widetilde{\nabla}^W \mathcal{L}(\theta) = \frac{1}{\epsilon} D(\theta)^{-1} \mathcal{U}_\theta(f^*), \tag{12}$$

*where $f^*$ is the unique solution to the quadratic optimization problem:*

$$\inf_{f \in \mathcal{H}} \mathcal{J}(f) := \int \|\nabla_x f(x)\|^2 \mathrm{d}\rho_\theta(x) + \frac{1}{\epsilon} \mathcal{U}_\theta(f)^\top D(\theta)^{-1} \mathcal{U}_\theta(f) + \lambda \|f\|_\mathcal{H}^2. \tag{13}$$

*When $\lambda = 0$, $f^*$ might not be well defined, still, we have: $\widetilde{\nabla}^W \mathcal{L}(\theta) = \lim_{j \to \infty} \frac{1}{\epsilon} D(\theta)^{-1} \mathcal{U}_\theta(f_j)$ for any limiting sequence of (13).*

Proposition 4 allows to compute the kernelized natural gradient directly, provided that the functional optimization (13) can be solved. This circumvents the direct computation and inversion of the metric as suggested by (11). In Section 3.2, we propose a method to efficiently compute an approximate solution to (13) using Nyström projections. We also show in Section 3.4 that restricting the space of functions to $\mathcal{H}$ can still lead to a good approximation of the WNG provided that $\mathcal{H}$ enjoys some denseness properties.

## 3.2 Nyström Methods for the Kerenalized Natural Gradient

We are interested now in finding an approximate solution to (13) which will allow to compute an estimator for the WNG using Proposition 4. Here we consider $N$ samples $(Z_n)_{1 \leq n \leq N}$ from the latent distribution $\nu$ which are used to produce $N$ samples $(X_n)_{1 \leq n \leq N}$ from $\rho_\theta$ using the map $h_\theta$, i.e., $X_n = h_\theta(Z_n)$. We also assume we have access to an estimate of the *Euclidean gradient* $\nabla \mathcal{L}(\theta)$ which is denoted by $\widehat{\nabla \mathcal{L}(\theta)}$. This allows to compute an empirical version of the cost function in (13),

$$\hat{\mathcal{J}}(f) := \frac{1}{N} \sum_{n=1}^{N} \|\nabla_x f(X_n)\|^2 + \frac{1}{\epsilon} \widehat{\mathcal{U}_\theta}(f)^\top D(\theta)^{-1} \widehat{\mathcal{U}_\theta}(f) + \lambda \|f\|_\mathcal{H}^2, \tag{14}$$

where $\widehat{\mathcal{U}_\theta}(f)$ is given by: $\widehat{\mathcal{U}_\theta}(f) = \widehat{\nabla \mathcal{L}(\theta)} + \frac{1}{N} \sum_{n=1}^{N} \nabla h_\theta(Z_n))^\top \nabla_x f(X_n)$. (14) has a similar structure as the empirical version of the kernel Sobolev distance introduced in Mroueh et al. (2019), it is also similar to another functional arising in the context of *score* estimation for *infinite dimensional exponential families* (Sriperumbudur et al., 2017; Sutherland et al., 2017; Arbel and Gretton, 2017). It can be shown using the generalized Representer Theorem (Schölkopf et al., 2001) that the optimal function minimizing (14) is a linear combination of functions of the form $x \mapsto \partial_i k(X_n, x)$ with $1 \leq n \leq N$ and $1 \leq i \leq d$ and $\partial_i k(y, x)$ denotes the partial derivative of $k$ w.r.t. $y_i$. This requires to solve a system of size $Nd \times Nd$ which can be prohibitive when both $N$ and $d$ are large. Nyström methods provide a way to improve such computational cost by further restricting the optimal solution to belong to a finite dimensional subspace $\mathcal{H}_M$ of $\mathcal{H}$ called the *Nyström subspace*. In the context of *score* estimation, Sutherland et al. (2017) proposed to use a subspace formed by linear combinations of the *basis functions* $x \mapsto \partial_i k(Y_m, x)$:

$$span\{x \mapsto \partial_i k(Y_m, x) \,|\, 1 \leq m \leq M; \quad 1 \leq i \leq d\}, \tag{15}$$

where $(Y_m)_{1 \leq m \leq M}$ are *basis points* drawn uniformly from $(X_n)_{1 \leq n \leq N}$ with $M \leq N$. This further reduces the computational cost when $M \ll N$ but still has a cubic dependence in the dimension $d$ since all partial derivatives of the kernel are considered to construct (15). Here, we propose to randomly sample one component of $(\partial_i k(Y_m, .))_{1 \leq i \leq d}$ for each basis point $Y_m$. Hence, we consider $M$ indices $(i_m)_{1 \leq m \leq M}$ uniformly drawn form $\{1, ..., d\}$ and define the *Nyström subspace* $\mathcal{H}_M$ to be:

$$\mathcal{H}_M := span\{x \mapsto \partial_{i_m} k(Y_m, x) | 1 \leq m \leq M\}.$$

An estimator for the *kernelized Wasserstein natural gradient* (KWNG) is then given by:

$$\widehat{\nabla^W \mathcal{L}(\theta)} = \frac{1}{\epsilon} D(\theta)^{-1} \widehat{\mathcal{U}_\theta}(\hat{f}^*), \qquad \hat{f}^* := \underset{f \in \mathcal{H}_M}{\operatorname{argmin}} \hat{\mathcal{J}}(f). \tag{16}$$

By definition of the Nyström subspace $\mathcal{H}_M$, the optimal solution $\hat{f}^*$ is necessarily of the form: $\hat{f}^*(x) = \sum_{m=1}^{M} \alpha_m \partial_{i_m} k(Y_m, x)$, where the coefficients $(\alpha_m)_{1 \leq m \leq M}$ are obtained by solving a finite dimensional quadratic optimization problem. This allows to provide a closed form expression for (17) in Proposition 5.

**Proposition 5.** *The estimator in (16) is given by:*

$$\widehat{\nabla^W \mathcal{L}(\theta)} = \frac{1}{\epsilon}\left( D(\theta)^{-1} - D(\theta)^{-1} T^\top \left( T D(\theta)^{-1} T^\top + \lambda \epsilon K + \frac{\epsilon}{N} C C^\top \right)^\dagger T D(\theta)^{-1} \right) \widehat{\nabla \mathcal{L}(\theta)}, \quad (17)$$

*where $C$ and $K$ are matrices in $\mathbb{R}^{M \times Nd}$ and $\mathbb{R}^{M \times M}$ given by*

$$C_{m,(n,i)} = \partial_{i_m} \partial_{i+d} k(Y_m, X_n), \qquad K_{m,m'} = \partial_{i_m} \partial_{i_{m'}+d} k(Y_m, Y_{m'}), \qquad (18)$$

*while $T$ is a matrix in $\mathbb{R}^{M \times q}$ obtained as the Jacobian of $\theta \mapsto \tau(\theta) \in \mathbb{R}^M$, i.e., $T := \nabla \tau(\theta)$, with*

$$(\tau(\theta))_m = \frac{1}{N} \sum_{n=1}^{N} \partial_{i_m} k(Y_m, h_\theta(Z_n)).$$

In (18), we used the notation $\partial_{i+d} k(y, x)$ for the partial derivative of $k$ w.r.t. $x_i$. A proof of Proposition 5 is provided in Appendix C.2 and relies on the reproducing property (10) and its generalization for partial derivatives of functions. The estimator in Proposition 5 is in fact a low rank approximation of the natural gradient obtained from the dual representation of the metric (9). While low-rank approximations for the Fisher-Rao natural gradient were considered in the context of variational inference and for a Gaussian variational posterior (Mishkin et al., 2018), (17) can be applied as a plug-in estimator for any family $\mathcal{P}_\Theta$ obtained as an implicit model. We next discuss a numerically stable expression of (17), its computational cost and the choice of the damping term in Section 3.3. We then provide asymptotic rates of convergence for (17) in Section 3.4.

### 3.3 PRACTICAL CONSIDERATIONS

**Numerically stable expression.** When $\lambda = 0$, the estimator in (17) has an additional structure which can be exploited to get more accurate solutions. By the chain rule, the matrix $T$ admits a second expression of the form $T = CB$ where $B$ is the Jacobian matrix of $(h_\theta(Z_n))_{1 \leq n \leq N}$. Although this expression is impractical to compute in general, it suggests that $C$ can be 'simplified'. This simplification can be achieved in practice by computing the SVD of $CC^\top = USU^T$ and pre-multiplying $T$ by $S^\dagger U^T$. The resulting expression is given in Proposition 6 and falls into the category of *Ridgeless estimators* (Liang and Rakhlin (2019)).

**Proposition 6.** *Consider an SVD decomposition of $CC^T$ of the form $CC^\top = USU^T$, then (17) is equal to:*

$$\widehat{\nabla^W \mathcal{L}(\theta)} = \frac{1}{\epsilon}\left( D(\theta)^{-1} - D(\theta)^{-1} \widetilde{T}^\top \left( \widetilde{T} D(\theta)^{-1} \widetilde{T}^\top + \frac{\epsilon}{N} P \right)^\dagger \widetilde{T} D(\theta)^{-1} \right) \widehat{\nabla \mathcal{L}(\theta)}, \qquad (19)$$

*where $P := S^\dagger S$ and* $\qquad \widetilde{T} := S^\dagger U^T T$.

**Choice of damping term.** So far, we only required $D(\theta)$ to be a diagonal matrix with positive coefficients. While a natural choice would be the identity matrix, this doesn't necessarily represent the best choice. As discussed by Martens and Sutskever (2012, Section 8.2), using the identity breaks the self-rescaling properties enjoyed by the natural gradient. Instead, we consider a scale-sensitive choice by setting $(D(\theta))_i = \|\widetilde{T}_{.,i}\|$ where $\widetilde{T}$ is defined in Proposition 6. When the sample-size is limited, as it is often the case when $N$ is the size of a mini-batch, larger values for $\epsilon$ might be required. That is to prevent the KWNG from over-estimating the step-size in low curvature directions. Indeed, these directions are rescaled by the inverse of the smallest eigenvalues of the information matrix which are harder to estimate accurately. To adjust $\epsilon$ dynamically during training, we use a variant of the Levenberg-Marquardt heuristic as in Martens and Sutskever (2012) which seems to perform well in practice; see Section 4.

**Computational cost.** The number of basis points $M$ controls the computational cost of both (17) and (19) which is dominated by the cost of computing $T$ and $C$, solving an $M \times M$ linear system and performing an SVD of $CC^T$ in the case of (19). This gives an overall cost of $O(dNM^2 + qM^2 + M^3)$. In practice, $M$ can be chosen to be small ($M \leq 20$) while $N$ corresponds to the number of samples in a mini-batch. Hence, in a typical deep learning model, most of the computational cost is due to computing $T$ as the typical number of parameters $q$ is of the order of millions. In fact, $T$ can be computed using automatic differentiation and would require performing $M$ backward passes on the model to compute the gradient for each component of $\tau$. Overall, the proposed estimator can be efficiently implemented and used for typical deep learning problems as shown in Section 4.

**Choice of the kernel.** We found that using either a gaussian kernel or a rational quadratic kernel to work well in practice. We also propose a simple heuristic to adapt the bandwidth of those kernels to the data by setting it to $\sigma = \sigma_0 \sigma_{N,M}$, where $\sigma_{N,M}$ is equal to the average square distance between samples $(X_n)_{1 \leq n \leq N}$ and the basis points $(Y_m)_{1 \leq m \leq M}$ and $\sigma_0$ is fixed a priori. Another choice is the median heuristic Garreau et al. (2018).

## 3.4 THEORY

In this section we are interested in the behavior of the estimator in the limit of large $N$ and $M$ and when $\lambda > 0$; we leave the case when $\lambda = 0$ for future work. We work under Assumptions (**A**) to (**G**) in Appendix A.2 which state that $\Omega$ is a non-empty subset, $k$ is continuously twice differentiable with bounded second derivatives, $\nabla h_\theta(z)$ has at most a linear growth in $z$ and $\nu$ satisfies some standard moments conditions. Finally, we assume that the estimator of the euclidean gradient $\widehat{\nabla \mathcal{L}(\theta)}$ satisfies Chebychev's concentration inequality which is often the case in Machine learning problem as discussed in Remark 1 of Appendix A.2. We distinguish two cases: the *well-specified* case and the *miss-specified* case. In the *well-specified* case, the vector valued functions $(\phi_i)_{1 \leq i \leq q}$ involved in Definition 2 are assumed to be gradients of functions in $\mathcal{H}$ and their smoothness is controlled by some parameter $\alpha \geq 0$ with worst case being $\alpha = 0$. Under this assumption, we obtain smoothness dependent convergence rates as shown in Theorem 14 of Appendix C.3 using techniques from Rudi et al. (2015); Sutherland et al. (2017). Here, we will only focus on the *miss-specified* which relies on a weaker assumption:

**Assumption 1.** *There exists two constants $C > 0$ and $c \geq 0$ such that for all $\kappa > 0$ and all $1 \leq i \leq q$, there is a function $f_i^\kappa$ satisfying:*

$$\|\phi_i - \nabla f_i^\kappa\|_{L_2(\rho_\theta)} \leq C\kappa, \qquad \|f_i^\kappa\|_{\mathcal{H}} \leq C\kappa^{-c}. \tag{20}$$

The left inequality in (20) represents the accuracy of the approximation of $\phi_i$ by gradients of functions in $\mathcal{H}$ while the right inequality represents the complexity of such approximation. Thus, the parameter $c$ characterizes the difficulty of the problem: a higher value of $c$ means that a more accurate approximation of $\phi_i$ comes at a higher cost in terms of its complexity. Theorem 7 provides convergences rates for the estimator in Proposition 5 under Assumption 1:

**Theorem 7.** *Let $\delta$ be such that $0 \leq \delta \leq 1$ and $b := \frac{1}{2+c}$. Under Assumption 1 and Assumptions (**A**) to (**G**) listed in Appendix A.2, for $N$ large enough, $M \sim (dN^{\frac{1}{2b+1}} \log(N))$, $\lambda \sim N^{\frac{1}{2b+1}}$ and $\epsilon \lesssim N^{-\frac{b}{2b+1}}$, it holds with probability at least $1 - \delta$ that:*

$$\|\widehat{\nabla^W \mathcal{L}(\theta)} - \nabla^W \mathcal{L}(\theta)\|^2 = \mathcal{O}\left(N^{-\frac{2}{4+c}}\right).$$

A proof of Theorem 7 is provided in Appendix C.3. In the best case where $c = 0$, we recover a convergence rate of $\frac{1}{\sqrt{N}}$ as in the *well specified* case for the worst smoothness parameter value $\alpha = 0$. Hence, Theorem 7 is a consistent extension of the *well-specified* case. For harder problems where $c > 0$ more basis points are needed, with $M$ required to be of order $dN\log(N)$ in the limit when $c \to \infty$ in which case the Nyström approximation loses its computational advantage.

## 4 EXPERIMENTS

This section presents an empirical evaluation of (KWNG) based on (19). Code for the experiments is available at https://github.com/MichaelArbel/KWNG.

## 4.1 CONVERGENCE ON SYNTHETIC MODELS

To empirically assess the accuracy of KWNG, we consider three choices for the parametric model $\mathcal{P}_\Theta$: the multivariate normal model, the multivariate log-normal model and uniform distributions on hyper-spheres. All have the advantage that the WNG can be computed in closed form (Chen and Li, 2018; Malagò et al., 2018). While the first models admit a density, the third one doesn't, hence the Fisher natural gradient is not defined in this case. While this choice of models is essential to obtain closed form expressions for WNG, the proposed estimator is agnostic to such choice of family. We also assume we have access to the exact Euclidean Gradient (EG) which is used to compute both of WNG and KWNG.

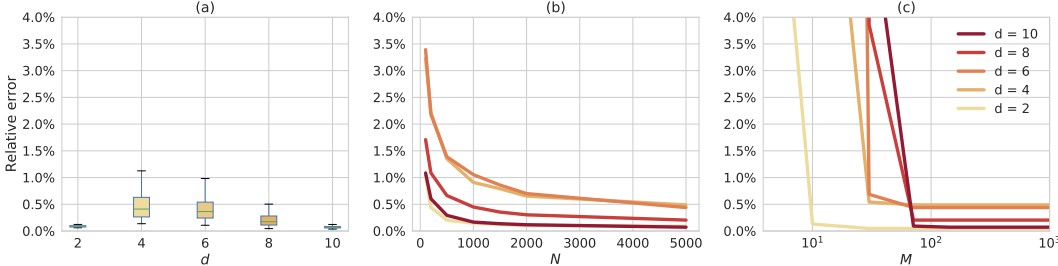

Figure 1: Relative error of KWNG averaged over 100 runs for varying dimension form $d=1$ (yellow) to $d=10$ (dark red) for the hyper-sphere model. (a): box-plot of the relative error as $d$ increases while $N=5000$ and $M=\lfloor d\sqrt{N}\rfloor$. (b) Relative error as the sample size $N$ increases and $M=\lfloor d\sqrt{N}\rfloor$. (c): Relative error as $M$ increases and $N=5000$. A gaussian kernel is used with a fixed bandwidth $\sigma=1$.

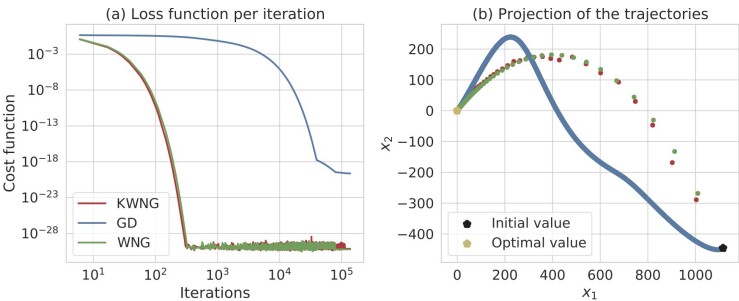

Figure 2: Left (a): Training error per iteration for KWNG, WNG, and EG. Right (b): projection of the sequence of updates obtained using KWNG, WNG and EG along the first two PCA directions of the WNG trajectory. The dimension of the sample space is fixed to $d=10$. Exact valued for the gradient are used for EG and WNG. For KWNG, $N=128$ samples and $M=100$ basis points are used. The regularization parameters are set to: $\lambda=0$ and $\epsilon=10^{-10}$. An optimal step-size $\gamma_t$ is used: $\gamma_t=0.1$ for both KWNG and WNG while $\gamma_t=0.0001$ for EG.

Figure 1 shows the evolution of the the relative error w.r.t. the sample-size $N$, the number of basis points $M$ and the dimension $d$ in the case of the hyper-sphere model. As expected from the consistency results provided in Section 3.4, the relative error decreases as the samples size $N$ increases. The behavior in the number of basis points $M$ shows a clear threshold beyond which the estimator becomes consistent and where increasing $M$ doesn't decrease the relative error anymore. This threshold increases with the dimension $d$ as discussed in Section 3.4. In practice, using the rule $M=\lfloor d\sqrt{N}\rfloor$ seems to be a good heuristic as shown in Figure 1 (a). All these observations persist in the case of the normal and log-normal model as shown in Figure 4 of Appendix D.1. In addition we report in Figure 5 the sensitivity to the choice of the bandwidth $\sigma$ which shows a robustness of the estimator to a wide choice of $\sigma$.

We also compare the optimization trajectory obtained using KWNG with the trajectories of both the exact WNG and EG in a simple setting: $\mathcal{P}_\Theta$ is the multivariate normal family and the loss function $\mathcal{L}(\theta)$ is the squared Wasserstein 2 distance between $\rho_\theta$ and a fixed target distribution $\rho_{\theta^*}$. Figure 2 (a), shows the evolution of the loss function at every iteration. There is a clear advantage of using the WNG over EG as larger step-sizes are allowed leading to faster convergence. Moreover, KWNG maintains this properties while being agnostic to the choice of the model. Figure 2 (b) shows the projected dynamics of the three methods along the two PCA directions of the WNG trajectory with highest variance. The dynamics of WNG seems to be well approximated by the one obtained using KWNG.

## 4.2 APPROXIMATE INVARIANCE TO PARAMETRIZATION

We illustrate now the approximate invariance to parametrization of the KWNG and show its benefits for training deep neural networks when the model is ill-conditioned. We consider a classification task on two datasets `Cifar10` and `Cifar100` with a Residual Network He et al. (2015). To use the KWNG

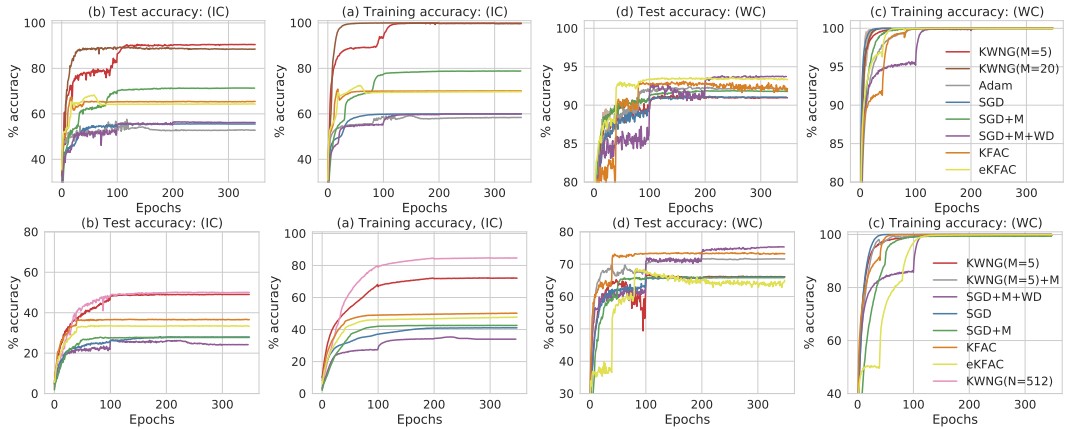

Figure 3: Test accuracy and Training accuracy for classification on `Cifar10` (top) and `Cifar100` (bottom) in both the ill-conditioned case (left side) and well-conditioned case (right side) for different optimization methods. on `Cifar10` Results are averaged over 5 independent runs except for KFAC and eKFAC.

estimator, we view the input RGB image as a latent variable $z$ with probability distribution $\nu$ and the output logits of the network $x := h_\theta(z)$ as a sample from the model distribution $\rho_\theta \in \mathcal{P}_\Theta$ where $\theta$ denotes the weights of the network. The loss function $\mathcal{L}$ is given by:

$$\mathcal{L}(\theta) := \int y(z)^\top \log(SM(Uh_\theta(z))) \mathrm{d}\nu(z),$$

where $SM$ is the Softmax function, $y(z)$ denotes the one-hot vector representing the class of the image $z$ and $U$ is a fixed invertible diagonal matrix which controls how well the model is conditioned. We consider two cases, the *Well-conditioned* case (WC) in which $U$ is the identity and the *Ill-conditioned* case (IC) where $U$ is chosen to have a condition number equal to $10^7$. We compare the performance of the proposed method with several variants of SGD: plain SGD, SGD + Momentum, and SGD + Momentum + Weight decay. We also compare with Adam Kingma and Ba (2014), KFAC optimizer (Martens and Grosse, 2015; Grosse and Martens, 2016) and eKFAC (George et al., 2018) which implements a fast approximation of the empirical Fisher Natural Gradient. We emphasize that gradient clipping by norm was used for all experiments and was crucial for a stable optimization using KWNG. Details of the experiments are provided in Appendix D.2. Figure 3 shows the training and test accuracy at each epoch on `Cifar10` in both (WC) and (IC) cases. While all methods achieve a similar test accuracy in the (WC) case on both datasets, methods based on the Euclidean gradient seem to suffer a drastic drop in performance in the (IC) case. This doesn't happen for KWNG (red line) which achieves a similar test accuracy as in (WC) case. Moreover, a speed-up in convergence in number of iterations can be obtained by increasing the number of basis points $M$ (brown line). The time cost is also in favor of KWNG (Figure 6). On `Cifar100`, KWNG is also less affected by the ill-conditioning, albeit to a lower extent. Indeed, the larger number of classes in `Cifar100` makes the estimation of KWNG harder as discussed in Section 4.1. In this case, increasing the batch-size can substantially improve the training accuracy (pink line). Moreover, methods that are used to improve optimization using the Euclidean gradient can also be used for KWNG. For instance, using Momentum leads to an improved performance in the (WC) case (grey line). Interestingly, KFAC seems to also suffer a drop in performance in the (IC) case. This might result from the use of an isotropic damping term $D(\theta) = I$ which would be harmful in this case. We also observe a drop in performance when a different choice of damping is used for KWNG. More importantly, using only a diagonal pre-conditioning of the gradient doesn't match the performance of KWNG (Figure 7).

ACKNOWLEDGEMENT

GM has received funding from the European Research Council (ERC) under the European Union's Horizon 2020 research and innovation programme (grant agreement nº 757983).

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

# A  PRELIMINARIES

## A.1  NOTATION

We recall that $\Omega$ is an open subset of $\mathbb{R}^d$ while $\Theta$ is an open subset of parameters in $\mathbb{R}^q$. Let $\mathcal{Z} \subset \mathbb{R}^p$ be a latent space endowed with a probability distribution $\nu$ over $\mathcal{Z}$. Additionally, $(\theta, z) \mapsto h_\theta(z) \in \Omega$ is a function defined over $\Theta \times \mathcal{Z}$. We consider a parametric set of probability distributions $\mathcal{P}_\Theta$ over $\Omega$ defined as the implicit model:

$$\mathcal{P}_\Theta := \{\rho_\theta := (h_\theta)_\# \nu \quad ; \quad \theta \in \Theta\},$$

where by definition, $\rho_\theta = (h_\theta)_\# \nu$ means that any sample $x$ from $\rho_\theta$ can be written as $x = h_\theta(z)$ where $z$ is a sample from $\nu$. We will write $B$ to denote the jacobian of $h_\theta$ w.r.t. $\theta$ viewed as a linear map from $\mathbb{R}^q$ to $L_2(\nu)^d$ without explicit reference to $\theta$:

$$Bu(z) = \nabla h_\theta(z).u; \qquad \forall u \in \mathbb{R}^q.$$

As in the main text, $\mathcal{L} : \Theta \to \mathbb{R}$ is a loss functions which is assumed to be of the form $\mathcal{L} = \mathcal{F}(\rho_\theta)$, with $\mathcal{F}$ being a real valued functional over the set of probability distributions. $\nabla \mathcal{L}(\theta)$ denotes the euclidean gradient of $\mathcal{L}$ w.r.t $\theta$ while $\widehat{\nabla \mathcal{L}(\theta)}$ is an estimator of $\nabla \mathcal{L}(\theta)$ using $N$ samples from $\rho_\theta$.

We also consider a Reproducing Kernel Hilbert Space $\mathcal{H}$ of functions defined over $\Omega$ with inner product $\langle .,. \rangle_\mathcal{H}$ and norm $\|.\|_\mathcal{H}$ and with a kernel $k : \Omega \times \Omega \to \mathbb{R}$. The reproducing property for the derivatives (Steinwart and Christmann, 2008, Lemma 4.34) will be important: $\partial_i f(x) = \langle f, \partial_i k(x,.) \rangle_\mathcal{H}$ for all $x \in \Omega$. It holds as long as $k$ is differentiable.

$C_b^\infty(\Omega)$ denotes the space of smooth bounded real valued functions on $\Omega$, and $C_c^\infty(\Omega) \subset C_b^\infty(\Omega)$ denotes the subset of compactly supported functions. For any measured space $\mathcal{Z}$ with probability distribution $\nu$, we denote by $L_2(\nu)$ the space of real valued and square integrable functions under $\nu$ and by $L_2(\nu)^d$ the space of square integrable vector valued functions under $\nu$ and with values in $\mathbb{R}^d$.

## A.2  ASSUMPTIONS

We make the following set of assumptions:

- **(A)** $\Omega$ is a non-empty open subset of $\mathbb{R}^d$.
- **(B)** There exists positive constants $\zeta$ and $\sigma$ such that $\int \|z\|^p \mathrm{d}\nu(z) \leq \frac{1}{2} p! \zeta^{p-2} \sigma^2$ for any $p \geq 2$.
- **(C)** For all $\theta \in \Theta$ there exists $C(\theta)$ such that $\|\nabla_\theta h_\theta(z)\| \leq C(\theta)(1 + \|z\|)$ for all $z \in \mathcal{Z}$.
- **(D)** $k$ is twice continuously differentiable on $\Omega \times \Omega$.
- **(E)** For all $\theta \in \Theta$ it holds that $\int \partial_i \partial_{i+d} k(x,x) \mathrm{d}p_\theta(x) < \infty$ for all $1 \leq i \leq d$.
- **(F)** The following quantity is finite: $\kappa^2 = \sup_{\substack{x \in \Omega \\ 1 \leq i \leq q}} \partial_i \partial_{i+q} k(x,x)$.
- **(G)** For all $0 \leq \delta \leq 1$, it holds with probability at least $1 - \delta$ that $\|\widehat{\nabla \mathcal{L}(\theta)} - \nabla \mathcal{L}(\theta)\| \lesssim N^{-\frac{1}{2}}$.

**Remark 1.** *Assumption (G) holds if for instance $\widehat{\nabla \mathcal{L}(\theta)}$ can be written as an empirical mean of i.i.d. terms with finite variance:*

$$\widehat{\nabla \mathcal{L}(\theta)} = \frac{1}{N} \sum_{i=1}^{N} \nabla_\theta l(h_\theta(Z_i))$$

*where $Z_i$ are i.i.d. samples from the latent distribution $\nu$ where $\int \nabla_\theta l(h_\theta(z)) \mathrm{d}\nu(z) = \mathcal{L}(\theta)$. This is often the case in the problems considered in machine-learning. In, this case, the sum of variances of the vector $\widehat{\nabla \mathcal{L}(\theta)}$ along its coordinates satisfies:*

$$\int \|\widehat{\nabla \mathcal{L}(\theta)} - \nabla \mathcal{L}(\theta)\|^2 \mathrm{d}\nu(z) = \frac{1}{N} \int \|\nabla_\theta l(h_\theta(z))\|^2 \mathrm{d}\nu(z) := \frac{1}{N} \sigma^2$$

*One can then conclude using Cauchy-Schwarz inequality followed by Chebychev's inequality that with probability $1 - \delta$:*

$$\|\widehat{\nabla \mathcal{L}(\theta)} - \nabla \mathcal{L}(\theta)\| \leq \frac{\sigma}{\sqrt{\delta N}}$$

*Moreover, Assumption (C) is often satisfied when the implicit model is chosen to be a deep networks with ReLU non-linearity.*

### A.3 Operators definition

**Differential operators.** We introduce the linear $L$ operator and its adjoint $L^\top$:

$$L : \mathcal{H} \to L_2(\nu)^d \qquad\qquad L^\top : L_2(\nu)^d \to \mathcal{H}$$

$$f \mapsto (\partial_i f \circ h_\theta)_{1 \le i \le d} \qquad\qquad v \mapsto \int \sum_{i=1}^d \partial_i k(h_\theta(z), .) v_i(z) \mathrm{d}\nu(z)$$

This allows to obtain the linear operator $A$ defined in Assumption 2 in the main text by composition $A := L^\top L$. We recall here another expression for $A$ in terms of outer product $\otimes$ and its regularized version for a given $\lambda > 0$,

$$A = \int \sum_{i=1}^d \partial_i k(h_\theta(z), .) \otimes \partial_i k(h_\theta(z), .) \mathrm{d}\nu(z) \qquad A_\lambda := A + \lambda I.$$

It is easy to see that $A$ is a symmetric positive operator. Moreover, it was established in Sriperumbudur et al. (2017) that $A$ is also a compact operator under Assumption (E).

Assume now we have access to $N$ samples $(Z_n)_{1 \le n \le N}$ as in the main text. We define the following objects:

$$\hat{A} := \frac{1}{N} \sum_{n=1}^N \sum_{i=1}^d \partial_i k(h_\theta(Z_n), .) \otimes \partial_i k(h_\theta(Z_n), .), \qquad \hat{A}_\lambda := \hat{A} + \lambda I.$$

Furthermore, if $v$ is a continuous function in $L_2(\nu)^d$, then we can also consider an empirical estimator for $L^\top v$:

$$\widehat{L^\top v} := \frac{1}{N} \sum_{n=1}^N \sum_{i=1}^d \partial_i k(h_\theta(Z_n), .) v_i(Z_n).$$

**Subsampling operators.** We consider the operator $Q_M$ defined from $\mathcal{H}$ to $\mathbb{R}^M$ by:

$$Q_M := \frac{\sqrt{q}}{\sqrt{M}} \sum_{m=1}^M e_m \otimes \partial_{i_m} k(Y_m, .) \tag{21}$$

where $(e_m)_{1 \le m \le M}$ is an orthonormal basis of $\mathbb{R}^M$. $Q_M$ admits a singular value decomposition of the form $Q_M = U \Sigma V^\top$, with $V V^\top := P_M$ being the orthogonal projection operator on the Nyström subspace $\mathcal{H}_M$. Similarly to Rudi et al. (2015); Sutherland et al. (2017), we define the projected inverse function $\mathcal{G}_M(C)$ as:

$$\mathcal{G}_M(C) = V(V^\top C V)^{-1} V^\top.$$

We recall here some properties of $\mathcal{G}_M$ from (Sutherland et al., 2017, Lemma 1):

**Lemma 8.** *Let $A : \mathcal{H} \to \mathcal{H}$ be a positive operator, and define $A_\lambda = A + \lambda I$ for any $\lambda > 0$. The following holds:*

1. $\mathcal{G}_M(A) P_M = \mathcal{G}_M(A)$

2. $P_M \mathcal{G}_M(A) = \mathcal{G}_M(A)$

3. $\mathcal{G}_M(A_\lambda) A_\lambda P_M = P_M$

4. $\mathcal{G}_M(A_\lambda) = (P_M A P_M + \lambda I)^{-1} P_M$

5. $\| A_\lambda^{\frac{1}{2}} \mathcal{G}_M(A_\lambda) A_\lambda^{\frac{1}{2}} \|$

**Estimators of the Wasserstein information matrix.** Here we would like to express the estimator in Proposition 5 in terms of the operators introduced previously. We have the following proposition:

**Proposition 9.** *The estimator defined in Proposition 5 admits the following representation:*

$$\widehat{\nabla^W \mathcal{L}(\theta)} = (\epsilon D(\theta) + G_{M,N})^{-1} \widehat{\nabla \mathcal{L}(\theta)}$$

*where $G_{M,N}$ is given by:*

$$G_{M,N} := (\widehat{L^\top B})^\top \mathcal{G}_M(\hat{A}_\lambda) \widehat{L^\top B}.$$

*Proof.* This a direct consequence of the minimax theorem (Ekeland and Témam, 1999, Proposition 2.3, Chapter VI) and applying (Sutherland et al., 2017, Lemma 3). □

The matrix $G_{M,N}$ is in fact an estimator of the Wasserstein information matrix defined in Definition 2. We will also need to consider the following population version of $G_{M,N}$ defined as :

$$G_M := (L^\top B)^\top \mathcal{G}_M(A_\lambda) L^\top B \tag{22}$$

# B    BACKGROUND IN INFORMATION GEOMETRY

## B.1    FISHER-RAO STATISTICAL MANIFOLD

In this section we briefly introduce the non-parametric Fisher-Rao metric defined over the set $\mathcal{P}$ of probability distributions with positive density. More details can be found in Holbrook et al. (2017). By abuse of notation, an element $\rho \in \mathcal{P}$ will be identified with its density which will also be denoted by $\rho$. Consider $\mathcal{T}_\rho$, the set of real valued functions $f$ defined over $\Omega$ and satisfying

$$\int \frac{f(x)^2}{\rho(x)} dx < \infty; \qquad \int f(x)\rho(x)dx = 0.$$

We have the following definition for the Fisher-Rao metric:

**Definition 4** (Fisher-Rao metric)**.** *The Fisher-Rao metric $g^F$ is defined for all $\rho \in \mathcal{P}$ as an inner product over $\mathcal{T}_\rho$ of the form:*

$$g_\rho^F(f,g) := \int \frac{1}{\rho(x)} f(x)g(x)dx, \qquad \forall f,g \in \mathcal{T}_\rho$$

Note that the choice of the set $\mathcal{T}_\rho$ is different from the one considered in Holbrook et al. (2017) which replaces the integrability condition by a smoothness one. In fact, it can be shown that these choices result in the same metric by a density argument.

## B.2    WASSERSTEIN STATISTICAL MANIFOLD

In this section we review the theory of Wasserstein statistical manifold introduced in Li and Montufar (2018a); Chen and Li (2018). By analogy to the Fisher-Rao metric which allows to endow the parametric model $\mathcal{P}_\Theta$ with the structure of a Riemannian manifold, it is also possible to use a different metric that is derived from the Wasserstein 2 distance. We first start by briefly introducing the Wasserstein 2 distance. Given two probability distributions $\rho$ and $\rho'$, we consider the set of all joint probability distributions $\Pi(\rho, \rho')$ between $\rho$ and $\rho'$ usually called the set of *couplings* between $\rho$ and $\rho'$. Any coupling $\pi$ defines a way of transporting mass from $\rho$ to $\rho'$. The cost of such transport can be measured as the expected distance between an element of mass of $\rho$ at location $x$ that is mapped to an element of mass of $\rho'$ at location $y$ using the coupling $\pi$:

$$\int \|x-y\|^2 d\pi(x,y)$$

The squared Wasserstein 2 distance between $\rho$ and $\rho'$ is defined as the smallest transport cost over all possible couplings:

$$W_2^2(\rho,\rho') = \inf_{\pi \in \Pi(\rho,\rho')} \int \|x-y\|^2 d\pi(x,y).$$

A dynamical formulation of $W_2$ was provided by the celebrated Benamou-Brenier formula in Benamou and Brenier (2000):

$$W_2^2(\rho,\rho') = \inf_{\phi_t} \int_0^1 \int \|\phi_l(x)\|^2 d\rho_l(x)dl$$

where the infimum is taken over the set of vector fields $\phi : [0,1] \times \Omega \to \mathbb{R}^d$. Each vector field of the potential determines a corresponding probability distribution $\rho_l$ as the solution of the continuity equation:

$$\partial_l \rho_l + div(\rho_l \phi_l) = 0, \qquad \rho_0 = \rho, \rho_1 = \rho'. \tag{23}$$

When $\Omega$ is a compact set, a Neumann condition is added on the boundary of $\Omega$ to ensure that the total mass is conserved. Such formulation suggests that $W_2(\rho,\rho')$ corresponds in fact to the shortest path from $\rho$ to $\rho'$. Indeed, given a path $\rho_l$ from $\rho$ to $\rho'$, the infinitesimal displacement direction is given by the distribution $\partial_l \rho_l$. The length $|\partial_l \rho_l|$ of this direction is measured by: $|\partial_l \rho_l|^2 := \int \|\phi_l(x)\|^2 \mathrm{d}\rho_l(x)$ Hence, $W_2^2(\rho,\rho')$ can be written as:

$$W_2^2(\rho\rho') = \inf_{\rho_l} \int_0^1 |\partial_l \rho_l|^2 \mathrm{d}l.$$

In fact, $\partial_l \rho_l$ can be seen as an element in the tangent space $T_{\rho_l} \mathcal{P}_2$ to $\mathcal{P}_2$ at point $\rho_l$. To ensure that (23) is well defined, $T_\rho \mathcal{P}_2$ can be defined as the set of distributions $\sigma$ satisfying $\sigma(1) = 0$.

$$|\sigma(f)| \leq C \|\nabla f\|_{L_2(\rho)}, \qquad \forall f \in C_c^\infty(\Omega) \tag{24}$$

for some positive constant $C$. Indeed, the condition in (24) guarantees the existence of a vector field $\phi_\sigma$ that is a solution to the PDE: $\sigma = -div(\rho \phi_\sigma)$.

Moreover, $|\partial_l \rho_l|^2$ can be seen as an inner product of $\partial_l \rho_l$ with itself in $T_{\rho_l} \mathcal{P}_2$. This inner product defines in turn a metric tensor $g^W$ on $\mathcal{P}_2$ called the Wasserstein metric tensor (see Otto and Villani (2000); Ambrosio et al. (2004)):

**Definition 5.** *The Wasserstein metric $g^W$ is defined for all $\rho \in \mathcal{P}_2$ as the inner product over $T_\rho \mathcal{P}_2$ of the form:*

$$g_\rho^W(\sigma,\sigma') := \int \phi_\sigma(x)^\top \phi_{\sigma'}(x) \mathrm{d}\rho(x), \qquad \forall \sigma,\sigma' \in T_\rho \mathcal{P}_2$$

*where $\phi_\sigma$ and $\phi_{\sigma'}$ are solutions to the partial differential equations:*

$$\sigma = -div(\rho \phi_\sigma), \qquad \sigma' = -div(\rho \phi_{\sigma'}).$$

*Moreover, $\phi_\sigma$ and $\phi_{\sigma'}$ are required to be in the closure of gradient of smooth and compactly supported functions w.r.t. $L_2(\rho)^d$.*

Definition 5 allows to endow $\mathcal{P}_2$ with a formal Riemannian structure with $W_2$ being its geodesic distance:

$$W_2^2(\rho,\rho') = \inf_{\rho_l} \int_0^1 g_{\rho_l}(\partial_l \rho_l, \partial_l \rho_l) \mathrm{d}l.$$

### B.3   NEGATIVE SOBOLEV DISTANCE AND LINEARIZATION OF THE WASSERSTEIN DISTANCE

To device the Wasserstein natural gradient, one can exploit a Taylor expansion of $W$ which is given in terms of the *Negative Sobolev distance* $\|\rho_{\theta+u} - \rho_\theta\|_{H^{-1}(\rho_\theta)}$ as done in Mroueh et al. (2019):

$$W_2^2(\rho_\theta,\rho_{\theta+u}) = \|\rho_{\theta+u} - \rho_\theta\|_{H^{-1}(\rho_\theta)}^2 + o(\|u\|^2).$$

Further performing a Taylor expansion of $\|\rho_{\theta+u} - \rho_\theta\|_{H^{-1}(\rho_\theta)}$ in $u$ leads to a quadratic term $u^\top G_W(\theta_t)u$ where we call $G_W(\theta_t)$ the *Wasserstein information matrix*. This two steps approach is convenient conceptually and allows to use the dual formulation of the *Negative Sobolev distance* to get an estimate of the quadratic term $u^\top G_W(\theta_t)u$ using kernel methods as proposed in Mroueh et al. (2019) for learning non-parametric models. However, with such approach, $\|\rho_{\theta+u} - \rho_\theta\|_{H^{-1}(\rho_\theta)}$ needs to be well defined for $u$ small enough. This requirement does not exploit the parametric nature of the problem and can be restrictive if $\rho_{\theta+u}$ and $\rho_\theta$ do not share the same support as we discuss now.

As shown in (Villani, 2003, Theorem 7.26) and discussed in (Arbel et al., 2019, Proposition 17 and 18), the Wasserstein distance between two probability distributions $\rho$ and $\rho'$ admits a first order expansion in terms of the Negative Sobolev Distance:

$$\lim_{\epsilon \to 0} \frac{1}{\epsilon} W_2(\rho,\rho+\epsilon(\rho'-\rho)) = \|\rho-\rho'\|_{H^{-1}(\rho)}$$

when $\rho'$ admits a bounded density w.r.t. $\rho$. When such assumption fails to hold, there are cases when this first order expansion is no longer available. For instance, in the simple case when the parametric family consists of dirac distributions $\delta_\theta$ located at a value $\theta$, the Wasserstein distance admits a closed form expression of the form:

$$W_2(\delta_\theta,\delta_\theta+\epsilon(\delta_{\theta'}-\delta_\theta)) = \sqrt{\epsilon}\|\theta-\theta'\|$$

Hence, $\frac{1}{\epsilon}W_2(\delta_\theta,\delta_\theta+\epsilon(\delta_{\theta'}-\delta_\theta))$ diverges to infinity. One can consider a different perturbation of the model $\delta_{\theta+\epsilon u}$ for some vector $u$ which the one we are interested in here. In this case, the Wasserstein distance admits a well-defined asymptotic behavior:

$$\lim_{\epsilon\to}\frac{1}{\epsilon}W_2(\delta_\theta,\delta_{\theta+\epsilon u})=\|u\|.$$

On the other hand the Negative Sobolev Distance is infinite for any value of $\epsilon$. To see this, we consider its dual formulation as in Mroueh et al. (2019):

$$\frac{1}{2}\|\rho-\rho'\|_{H^{-1}(\rho)}=\sup_{\substack{f\in C_c^\infty(\Omega)\\ \int f(x)\mathrm{d}\rho(x)=0}}\int f(x)\mathrm{d}\rho(x)-\int f(x)\mathrm{d}\rho'(x)-\frac{1}{2}\int\|\nabla_x f(x)\|^2\mathrm{d}\rho(x)$$

Evaluating this expression for $\delta_\theta$ and $\delta_{\theta+\epsilon u}$ for any value $\epsilon>0$ and for any $u$ that is non zero, on has:

$$\frac{1}{2\epsilon}\|\delta_\theta-\delta_{\theta+\epsilon u}\|_{H^{-1}(\delta_\theta)}=\sup_{\substack{f\in C_c^\infty(\Omega)\\ f(\theta)=0}}\frac{1}{\epsilon}(f(\theta)-f(\theta+\epsilon u))-\frac{1}{2}\|\nabla_x f(\theta)\|^2 \tag{25}$$

One can always find a function $f$ such that $\nabla f(\theta)=0$, $f(\theta)=0$ and $-f(\theta+\epsilon u)$ can be arbitrarily large, thus the Negative Sobolev distance is infinite. This is not the case of the metric $u^\top G_W(\theta)u$ which can be computed in closed form:

$$\frac{1}{2}u^\top G_W(\theta)u=\sup_{\substack{f\in C_c^\infty(\Omega)\\ f(\theta)=0}}\nabla f(\theta)^\top u-\frac{1}{2}\|\nabla_x f(\theta)\|^2 \tag{26}$$

In this case, choosing $f(\theta)=0$ and $\nabla f(\theta)=u$ achieves the supremum which is simply given by $\frac{1}{2}\|u\|^2$. Equation (26) can be seen as a limit case of (25) when $\epsilon\to 0$:

$$\frac{1}{2}u^\top G_W(\theta)u:=\sup_{\substack{f\in C_c^\infty(\Omega)\\ f(\theta)=0}}\lim_{\epsilon\to 0}\frac{1}{\epsilon}(f(\theta)-f(\theta+\epsilon u))-\frac{1}{2}\|\nabla_x f(\theta)\|^2$$

However, the order between the supremum and the limit cannot be exchanged in this case, which makes the two objects behave very differently in the case of singular probability distributions.

## C  PROOFS

### C.1  PRELIMINARY RESULTS

Here we provide a proof of the invariance properties of the Fisher and Wasserstein natural gradient descent in the continuous-time limit as stated in Proposition 1. Consider an invertible and smoothly differentiable re-parametrization $\Psi$, satisfying $\psi=\Psi(\theta)$. Denote by $\bar\rho_\psi=\rho_{\Psi^{-1}(\psi)}$ the re-parametrized model and $\bar G_W(\psi)$ and $\bar G_F(\psi)$ their corresponding Wasserstein and Fisher information matrices whenever they are well defined.

*Proof of Proposition 1.* Here we only consider the case when $\nabla^D\mathcal{L}(\theta)$ is either given by the Fisher natural gradient $\nabla^F\mathcal{L}(\theta)$ or the Wasserstein Natural gradient $\nabla^W\mathcal{L}(\theta)$. We will first define $\widetilde\psi_s:=\Psi(\theta_s)$ and show that in fact $\widetilde\psi_s=\psi_s$ at all times $s>0$. First, let's differentiate $\widetilde\psi_s$ in time:

$$\dot{\widetilde\psi}_s=-\nabla_\theta\Psi(\theta_s)^\top G_D(\theta_s)^{-1}\nabla_\theta\mathcal{L}(\theta_s)$$

By the chain rule, we have that $\nabla_\theta\mathcal{L}(\theta_s)=\nabla_\theta\Psi(\theta_s)\nabla_\psi\bar{\mathcal{L}}(\widetilde\psi_s)$, hence:

$$\dot{\widetilde\psi}_s=-\nabla_\theta\Psi(\theta_s)^\top G_D(\theta_s)^{-1}\nabla_\theta\Psi(\theta_s)\nabla_\psi\bar{\mathcal{L}}(\widetilde\psi_s).$$

It is easy to see that $\nabla_\theta\Psi^{-1}(\widetilde\psi_s)=(\nabla_\psi\Psi^{-1}(\psi_s))^{-1}$ by definition of $\Psi$ and $\widetilde\psi_s$, hence by Lemma 10 one can conclude that:

$$\dot{\widetilde\psi}_s=-G_D(\widetilde\psi_s)^{-1}\nabla_\psi\bar{\mathcal{L}}(\widetilde\psi_s).$$

Hence, $\widetilde\psi_s$ satisfies the same differential equation as $\psi_s$. Now keeping in mind that $\psi_0=\widetilde\psi_0=\Psi(\theta_0)$, it follows that $\psi_0=\widetilde\psi_0=\Psi(\theta_0)$ by uniqueness of differential equations. □

**Lemma 10.** *Under conditions of Propositions 2 and 3, the informations matrices $\bar{G}_W(\psi)$ and $\bar{G}_F(\psi)$ are related to $G_W(\theta)$ and $G_F(\theta)$ by the relation:*

$$\bar{G}_W(\psi) = \nabla_\psi \Psi^{-1}(\psi)^\top G_W(\theta) \nabla_\psi \Psi^{-1}(\psi)$$
$$\bar{G}_F(\psi) = \nabla_\psi \Psi^{-1}(\psi)^\top G_F(\theta) \nabla_\psi \Psi^{-1}(\psi)$$

*Proof.* Let $v \in R^q$ and write $u = \nabla_\theta \Psi^{-1}(\psi)v$, then by the dual formulations of $G_W(\theta)$ and $G_F(\theta)$ in Proposition 2 we have that:

$$\frac{1}{2} v^\top \nabla_\psi \Psi^{-1}(\psi)^\top G_F(\theta) \nabla_\psi \Psi^{-1}(\psi)v = \sup_{\substack{f \in C_c^\infty(\Omega) \\ \int f(x)\mathrm{d}\rho_\theta(x)=0}} \nabla \rho_\theta(f)^\top \nabla_\theta \Psi^{-1}(\psi)v - \frac{1}{2}\int f(x)^2 \mathrm{d}\rho_\theta(x)\mathrm{d}x,$$

Now recalling that $\nabla_\psi \bar{\rho}_\psi = \nabla_\theta \rho_\theta \nabla_\psi \Psi^{-1}(\psi)$ by Lemma 11, it follows that:

$$\frac{1}{2} v^\top \nabla_\psi \Psi^{-1}(\psi)^\top G_F(\theta) \nabla_\psi \Psi^{-1}(\psi)v = \sup_{\substack{f \in C_c^\infty(\Omega) \\ \int f(x)\mathrm{d}\rho_\theta(x)=0}} \nabla_\psi \bar{\rho}_\psi(f)^\top v - \frac{1}{2}\int f(x)^2 \mathrm{d}\rho_\theta(x)\mathrm{d}x,$$

Using again Proposition 2 for the reparametrized model $\bar{\rho}_\psi$, we directly have that:

$$\frac{1}{2} v^\top G_F(\psi)v = \sup_{\substack{f \in C_c^\infty(\Omega) \\ \int f(x)\mathrm{d}\rho_\theta(x)=0}} \nabla_\psi \bar{\rho}_\psi(f)^\top v - \frac{1}{2}\int f(x)^2 \mathrm{d}\rho_\theta(x)\mathrm{d}x,$$

The result follows by equating both expression. The same procedure can be applied for the case the Wasserstein information matrix using Proposition 3. $\qquad\square$

**Lemma 11.** *The distributional gradients $\nabla_\psi \bar{\rho}_\psi$ and $\nabla_\theta \rho_\theta$ are related by the expression:*

$$\nabla_\psi \bar{\rho}_\psi = \nabla_\theta \rho_\theta \nabla_\psi \Psi^{-1}(\psi)$$

*Proof.* The proof follows by considering a fixed direction $u \in \mathbb{R}^q$ and a test function $f \in \mathcal{C}_c^\infty(\Omega)$ and the definition of distributional gradient in Definition 3:

$$\nabla \bar{\rho}_\psi(f)^\top u = \lim_{\epsilon \to 0} \frac{1}{\epsilon} \int f(x)\mathrm{d}\bar{\rho}_{\psi+\epsilon u}(x) - \int f(x)\mathrm{d}\bar{\rho}_\psi(x)$$

$$= \lim_{\epsilon \to 0} \frac{1}{\epsilon} \int f(x)\mathrm{d}\rho_{\Psi^{-1}(\psi+\epsilon u)}(x) - \int f(x)\mathrm{d}\rho_{\Psi^{-1}(\psi)}(x)$$

Now by differentiability of $\Psi^{-1}$, we have the following first order expansion:

$$\Psi^{-1}(\psi+\epsilon u) = \Psi^{-1}(\psi) + \epsilon \nabla \Psi^{-1}(\psi)^\top u + \epsilon \upsilon(\epsilon)$$

where $\upsilon(\epsilon)$ converges to 0 when $\epsilon \to 0$. Now using again the definition Definition 3 for $\rho_{\Psi\psi}$ one has:

$$\frac{1}{\epsilon}\left( \int f(x)\mathrm{d}\rho_{\Psi^{-1}(\psi+\epsilon u)}(x) - \int f(x)\mathrm{d}\rho_{\Psi^{-1}(\psi)}(x) \right) = \nabla \rho_{\Psi^{-1}(\psi)}(f)^\top \nabla \Psi^{-1}(\psi)^\top u$$

$$+ \epsilon \upsilon(\epsilon) + \delta(\epsilon, f, (\nabla \Psi^{-1}(\psi)u + \upsilon(\epsilon)))$$

The last two terms converge to 0 as $\epsilon \to 0$, hence leading to the desired expression. $\qquad\square$

**Proposition 12.** *When $\rho_\theta$ admits a density that is continuously differentiable w.r.t $\theta$ and such that $x \mapsto \nabla \rho_\theta(x)$ is continuous, then the distributional gradient is of the form:*

$$\nabla \rho_\theta(f) = \int f(x)\nabla \rho_\theta(x)\mathrm{d}x, \qquad \forall f \in \mathcal{C}_c^\infty(\Omega)$$

*where $\nabla \rho_\theta(x)$ denotes the gradient of the density of $\rho_\theta(x)$ at $x$.*

*Proof.* Let $\epsilon > 0$ and $u \in \mathbb{R}^q$, we define the function $\nu(\epsilon, u, f)$ as follows:

$$\nu(\epsilon, u, f) = \int f(x) \left( \frac{1}{\epsilon} (\rho_{\theta + \epsilon u}) - \rho_\theta - \nabla \rho_\theta^\top u \right) \mathrm{d}x$$

we just need to show that $\nu(\epsilon, u, f) \to 0$ as $\epsilon \to 0$. This follows form the differentiation lemma (Klenke, 2008, Theorem 6.28) applied to the function $(\theta, x) \mapsto f(x) \rho_\theta(x)$. Indeed, this function is integrable in $x$ for any $\theta'$ in a neighborhood $U$ of $\theta$ that is small enough, it is also differentiable on that neighborhood $U$ and satisfies the domination inequality:

$$|f(x) \nabla \rho_\theta(x)^\top u| \leq |f(x)| \sup_{x \in Supp(f), \theta \in U} \nabla \rho_\theta(x)^\top u|.$$

The inequality follows from continuity of $(\theta, x) \nabla \rho_\theta(x)$ and recalling that $f$ is compactly supported. This concludes the proof. $\qquad \square$

We fist provide a proof of the dual formulation for the Fisher information matrix.

*Proof of Proposition 2 .* Consider the optimization problem:

$$\sup_{\substack{f \in C_c^\infty(\Omega) \\ \int f(x) \mathrm{d}\rho_\theta(x) = 0}} \left( \int f(x) \nabla \rho_\theta(x) \mathrm{d}x \right)^\top u - \frac{1}{2} \int f(x)^2 \rho_\theta(x) \mathrm{d}x \tag{27}$$

Recalling that the set of smooth and compactly supported functions $C_c^\infty(\infty)$ is dense in $L_2(\rho_\theta)$ and that the objective function in (27) is continuous and coercive in $f$, it follows that (27) admits a unique solution $f^*$ in $L_2(\rho_\theta)$ which satisfies the optimality condition:

$$\int f(x) (\nabla \rho_\theta(x))^\top u \mathrm{d}x = \int f(x) f^*(x) \rho_\theta(x) \mathrm{d}x \qquad \forall f \in L_2(\rho_\theta)$$

Hence, it is easy to see that $f^* = (\nabla \rho_\theta)^\top u / \rho_\theta$ and that the optimal value of (27) is given by:

$$\frac{1}{2} \int \frac{((\nabla \rho_\theta(x))^\top u)^2}{\rho_\theta(x)} \mathrm{d}x.$$

This is equal to $u^\top G_F(\theta) u$ by Definition 1. $\qquad \square$

The next proposition ensures that the Wasserstein information matrix defined in Definition 2 is well-defined and has a dual formulation.

**Proposition 13.** *Consider the model defined in (6) and let $(e_s)_{1 \leq s \leq q}$ be an orthonormal basis of $\mathbb{R}^q$. Under Assumptions (B) and (C), there exists an optimal solution $\Phi = (\phi_s)_{1 \leq s \leq q}$ with $\phi_s$ in $L_2(\rho_\theta)^d$ satisfying the PDE:*

$$\partial_s \rho_\theta = -div(\rho_\theta \phi_s)$$

*The elliptic equations also imply that $L^\top \nabla h_\theta = L^\top (\Phi \circ h_\theta)$. Moreover, the Wasserstein information matrix $G_W(\theta)$ on $\mathcal{P}_\Theta$ at point $\theta$ can be written as $G_W(\theta) = \Phi^\top \Phi$ where the inner-product is in $L_2(\rho_\theta)^d$ and satisfies:*

$$\frac{1}{2} u^\top G_W(\theta) u = \sup_{\substack{f \in C_c^\infty(\Omega) \\ \int f(x) \mathrm{d}\rho_\theta(x) = 0}} \nabla \rho_\theta(f)^\top u - \frac{1}{2} \int \|\nabla_x f(h_\theta(z))\|^2 \mathrm{d}\nu(z).$$

*for all $u \in \mathbb{R}^q$.*

*Proof.* Let $(e_s)_{1 \leq s \leq q}$ be an orthonormal basis of $\mathbb{R}^q$. For all $1 \leq s \leq q$, we will establish the existence of an optimal solution $\phi_s$ in $L_2(\rho_\theta)^d$ satisfying the PDE:

$$\partial_s \rho_\theta = -div(\rho_\theta \phi_s) \tag{28}$$

Consider the variational problem:

$$\sup_{\phi \in \mathcal{S}} \int \phi(h_\theta(z))^\top \partial_{\theta_s} h_\theta(z) - \frac{1}{2}\|\phi\|^2_{L_2(\rho_\theta)} \tag{29}$$

where $\mathcal{S}$ is a Hilbert space obtained as the closure in $L_2(\rho_\theta)^d$ of functions of the form $\phi = \nabla_x f$ with $f \in C_c^\infty(\Omega)$:

$$\mathcal{S} := \overline{\{\nabla_x f \mid f \in C_c^\infty(\Omega)\}}_{L_2(\rho_\theta^d)}.$$

We have by Assumption (C) that:

$$\int \phi(h_\theta(z))^\top \partial_{\theta_s} h_\theta(z) \mathrm{d}\nu(z) \leq \leq C(\theta)\sqrt{\int (1+\|z\|^2)\mathrm{d}\nu(z)} \int \|\phi\|_{L_2(\rho_\theta)}.$$

Moreover, by Assumption (B), we know that $\sqrt{\int(1+\|z\|^2)\mathrm{d}\nu(z)} < \infty$. This implies that the objective in (29) is continuous in $\phi$ while also being convex.s It follows that (29) admits a unique solution $\phi_s^* \in \mathcal{S}$ which satisfies for all $\phi \in \mathcal{S}$:

$$\int \phi(h_\theta(z))^\top \phi_s^*(h_\theta(z))\mathrm{d}\nu(z) = \int \phi(h_\theta(z))^\top \partial_{\theta_s} h_\theta(z))\mathrm{d}\nu(z)$$

In particular, for any $f \in C_c^\infty(\Omega)$, it holds that:

$$\int \nabla_x f(h_\theta(z))^\top \phi_s^*(h_\theta(z))\mathrm{d}\nu(z) = \int \nabla_x f(h_\theta(z))^\top \partial_{\theta_s} h_\theta(z))\mathrm{d}\nu(z)$$

which is equivalent to (28) and implies directly that $L^T \nabla h_\theta = L^T \Phi \circ h_\theta$ where $\Phi := (\phi_s^*)_{1 \leq s \leq q}$. The variational expression for $\frac{1}{2}u^\top G_W u$ follows by noting that (29) admits the same optimal value as

$$\sup_{\substack{f \in C_c^\infty(\Omega) \\ \int f(x)\mathrm{d}\rho_\theta(x)=0}} \nabla \rho_\theta(f)^\top u - \frac{1}{2}\int \|\nabla_x f(h_\theta(z))\|^2 \mathrm{d}\nu(z).$$

That is because $\mathcal{S}$ is by definition the closure in $L_2(\rho_\theta)^d$ of the set of gradients of smooth and compactly supported functions on $\Omega$. □

*Proof of Proposition 3.* This is a consequence of Proposition 13. □

## C.2 EXPRESSION OF THE ESTIMATOR

We provide here a proof of Proposition 5

*Proof of Proposition 5.* Here, to simplify notations, we simply write $D$ instead of $D(\theta)$. First consider the following optimization problem:

$$\inf_{f \in \mathcal{H}_M} \frac{1}{N}\sum_{n=1}^N \|\nabla f(X_n)\|^2 + \lambda\|f\|_{\mathcal{H}}^2 + \frac{1}{\epsilon}\mathcal{R}(f)^\top D^{-1}\mathcal{R}(f) + \frac{2}{\epsilon}\mathcal{R}(f)^\top D^{-1}\widehat{\nabla\mathcal{L}(\theta)}$$

with $\mathcal{R}(f)$ given by $\mathcal{R}(f) = \frac{1}{N}\sum_{n=1}^N \nabla f(X_n)^\top B(Z_n)$. Now, recalling that any $f \in \mathcal{H}_M$ can be written as $f = \sum_{m=1}^M \alpha_m \partial_{i_m} k(Y_m, .)$, and using the reproducing property $\partial_i f(x) = \langle f, \partial_i k(x, .)\rangle_{\mathcal{H}}$ (Steinwart and Christmann, 2008, Lemma 4.34) , it is easy to see that:

$$\frac{1}{N}\sum_{n=1}^N \|\nabla f(X_n)\|^2 = \frac{1}{N}\sum_{\substack{1 \leq n \leq N \\ 1 \leq i \leq d}} (\sum_{m=1}^M \alpha_m \partial_{i_m}\partial_{i+d}k(Y_m, X_n))^2.$$

$$\|f\|_{\mathcal{H}}^2 = \sum_{1 \leq m,m' \leq M} \alpha_m \alpha_{m'} \partial_{i_m}\partial_{i_{m'}+d}k(Y_m, Y_{m'})$$

$$\mathcal{R}(f) = \frac{1}{N}\sum_{\substack{1 \leq n \leq N \\ 1 \leq i \leq d \\ 1 \leq m \leq M}} \alpha_m \partial_{i_m}\partial_{i+d}k(Y_m, X_n)B_i(Z_n)$$

The above can be expressed in matrix form using the matrices defined in Proposition 5:

$$\frac{1}{N}\sum_{n=1}^{N}\|\nabla f(X_n)\|^2 = \alpha^\top CC^\top \alpha; \qquad \|f\|_\mathcal{H}^2 = \alpha^\top K\alpha; \qquad \mathcal{R}(f) = \alpha^\top CB.$$

Hence the optimal solution $\hat{f}^*$ is of the form $\hat{f}^* = \sum_{m=1}^{M}\alpha_m^*\partial_{i_m}k(Y_m,.)$, with $\alpha^*$ obtained as a solution to the finite dimensional problem in $\mathbb{R}^M$:

$$\min_{\alpha \in \mathbb{R}^M} \alpha^\top(\epsilon CC^\top + \epsilon\lambda K + CBD^{-1}B^\top C^\top)\alpha + 2\alpha^\top CBD^{-1}\widehat{\nabla\mathcal{L}(\theta)}$$

It is easy to see that $\alpha^*$ are given by:

$$\alpha^* = -(\epsilon CC^T + \epsilon\lambda K + CBD^{-1}B^TC^T)^\dagger CBD^{-1}\widehat{\nabla\mathcal{L}(\theta)}.$$

Now recall that the estimator in Proposition 5 is given by: $\widehat{\nabla^W\mathcal{L}(\theta)} = \frac{1}{\epsilon}D^{-1}\mathcal{U}_\theta(\hat{f}^*)$. Hence, $\frac{1}{\epsilon}D^{-1}(\widehat{\nabla\mathcal{L}(\theta)} - B^TC^T\alpha^*)$ The desired expression is obtained by noting that $CB = T$ using the chain rule.

$\square$

### C.3 CONSISTENCY RESULTS

**Well-specified case.** Here, we assume that the vector valued functions $(\phi_i)_{1\leq i\leq q}$ involved in Definition 2 can be expressed as gradients of functions in $\mathcal{H}$. More precisely:

**Assumption 2.** *For all $1 \leq i \leq q$, there exits functions $f_i \in \mathcal{H}$ such that $\phi_i = \nabla f_i$. Additionally, $f_i$ are of the form $f_i = A^\alpha v_i$ for some fixed $\alpha \geq 0$, with $v_i \in \mathcal{H}$ and $A$ being the differential covariance operator defined on $\mathcal{H}$ by $A: f \mapsto \int \sum_{i=1}^{d}\partial_i k(h_\theta(z),.)\partial_i f(h_\theta(z))\mathrm{d}\nu(z)$.*

The parameter $\alpha$ characterizes the smoothness of $f_i$ and therefore controls the statistical complexity of the estimation problem. Using a similar analysis as Sutherland et al. (2017) we obtain a convergence rate for the estimator in Proposition 5

the following convergence rates for the estimator in Proposition 5:

**Theorem 14.** *Let $\delta$ be such that $0 \leq \delta \leq 1$ and $b := \min(1, \alpha + \frac{1}{2})$. Under Assumption 2 and Assumptions (A) to (G) listed in Appendix A.2, for $N$ large enough, $M \sim (dN^{\frac{1}{2b+1}}\log(N))$, $\lambda \sim N^{-\frac{1}{2b+1}}$ and $\epsilon \lesssim N^{-\frac{b}{2b+1}}$, it holds with probability at least $1-\delta$ that:*

$$\|\widehat{\nabla^W\mathcal{L}(\theta)} - \nabla^W\mathcal{L}(\theta)\|^2 = \mathcal{O}\left(N^{-\frac{2b}{2b+1}}\right).$$

In the worst case where $\alpha = 0$, the proposed estimator needs at most $M \sim (d\sqrt{N}\log(N))$ to achieve a convergence rate of $N^{-\frac{1}{2}}$. The smoothest case requires only $M \sim (dN^{\frac{1}{3}}\log(N))$ to achieve a rate of $N^{-\frac{2}{3}}$. Thus, the proposed estimator enjoys the same statistical properties as the ones proposed by Sriperumbudur et al. (2017); Sutherland et al. (2017) while maintaining a computational advantage[1] tNow we provide a proof for Theorem 14 which relies on the same techniques used by Rudi et al. (2015); Sutherland et al. (2017).

*Proof of Theorem 14 .* The proof is a direct consequence of Proposition 15 under Assumption 2. $\square$

*Proof of Theorem 7.* The proof is a direct consequence of Proposition 15 under Assumption 1. $\square$

**Proposition 15.** *Under Assumptions (A) to (G) and for $0 \leq \delta \leq 1$ and $N$ large enough, it holds with probability at least $1-\delta$:*

$$\|\widehat{\nabla^W\mathcal{L}} - \nabla^W\mathcal{L}\| = \mathcal{O}(N^{-\frac{b}{2b+1}})$$

*provided that $M \sim dN^{\frac{1}{2b+1}}\log N$, $\lambda \sim N^{\frac{1}{2b+1}}$ and $\epsilon \lesssim N^{-\frac{b}{2b+1}}$ where $b := \min(1, \alpha + \frac{1}{2})$ when Assumption 2 holds and $b = \frac{1}{2+c}$ when Assumption 1 holds instead.*

---

[1] The estimator proposed by Sutherland et al. (2017) also requires $M$ to grow linearly with the dimension $d$ although such dependence doesn't appear explicitly in the statement of Sutherland et al. 2017, Theorem 2.

*Proof.* Here for simplicity we assume that $D(\theta) = I$ without loss of generality and we omit the dependence in $\theta$ and write $\nabla^W \mathcal{L}$ and $\nabla \mathcal{L}$ instead of $\nabla^W \mathcal{L}(\theta)$ and $\nabla \mathcal{L}(\theta)$ and $\nabla^W \mathcal{L}(\theta)$. We also define $\hat{G}_\epsilon = \epsilon I + G_{M,N}$ and $G_\epsilon = \epsilon I + G_W$. By Proposition 9, we know that $\widehat{\nabla^W \mathcal{L}} = \hat{G}_\epsilon^{-1} \widehat{\nabla \mathcal{L}}$. We use the following decomposition:

$$\|\widehat{\nabla^W \mathcal{L}} - \nabla^W \mathcal{L}\| \le \|\hat{G}_\epsilon^{-1}(\widehat{\nabla \mathcal{L}} - \nabla \mathcal{L})\| + \|\hat{G}_\epsilon^{-1}(G_{M,N} - G_W)G_W^{-1}\nabla \mathcal{L}\| + \epsilon \|\hat{G}_\epsilon^{-1} G_W^{-1} \nabla \mathcal{L}\|$$

To control the norm of $\hat{G}_\epsilon^{-1}$ we write $\hat{G}_\epsilon^{-1} = G_\epsilon^{-\frac{1}{2}}(H + I)^{-1} G_\epsilon^{-\frac{1}{2}}$, where $H$ is given by $H := G_\epsilon^{-\frac{1}{2}}(G_{M,N} - G_W)G_\epsilon^{-\frac{1}{2}}$. Hence, provided that $\mu := \lambda_{\max}(H)$, the highest eigenvalue of $H$, is smaller than 1, it holds that:

$$\|(H + I)^{-1}\| \le (1 - \mu)^{-1}.$$

Moreover, since $G_W$ is positive definite, its smallest eigenvalue $\eta$ is strictly positive. Hence, $\|G_\epsilon^{-1}\| \le (\eta + \epsilon)^{-1}$. Therefore, we have $\|\hat{G}_\epsilon^{-1}\| \le (\eta + \epsilon)(1 - \mu))^{-1}$, which implies:

$$\|\widehat{\nabla^W \mathcal{L}} - \nabla^W \mathcal{L}\| \le (\eta + \epsilon)^{-1}\left(\frac{\|\widehat{\nabla \mathcal{L}} - \nabla \mathcal{L}\|}{1 - \mu} + \eta^{-1}\|\nabla \mathcal{L}\|\|G_{M,N} - G_W\| + \epsilon \eta^{-1}\|\nabla \mathcal{L}\|\right).$$

Let $0 \le \delta \le 1$. We have by Assumption (G) that $\|\widehat{\nabla \mathcal{L}} - \nabla \mathcal{L}\| = \mathcal{O}(N^{-\frac{1}{2}})$ with probability at least $1 - \delta$. Similarly, by Proposition 16 and for $N$ large enough, we have with probability at least $1 - \delta$ that $\|G_{M,N} - G_W\| = \mathcal{O}(N^{-\frac{b}{2b+1}})$ where $b$ is defined in Proposition 16. Moreover, for $N$ large enough, one can ensure that $\mu \le \frac{1}{2}$ so that the following bound holds with probability at least $1 - \delta$:

$$\|\widehat{\nabla^W \mathcal{L}} - \nabla^W \mathcal{L}\| \lesssim (\eta + \epsilon)^{-1}\left(2N^{-\frac{1}{2}} + \eta^{-1}\|\nabla \mathcal{L}\|(N^{-\frac{b}{2b+1}} + \epsilon)\right).$$

Thus by setting $\epsilon \lesssim N^{-\frac{b}{2b+1}}$ we get the desired convergence rate. $\qquad\square$

**Proposition 16.** *For any $0 \le \delta \le 1$, we have with probability as least $1 - \delta$ and for $N$ large enough that:*

$$\|G_{M,N} - G_W\| = \mathcal{O}(N^{-\frac{b}{2b+1}}).$$

*provided that $M \sim dN^{\frac{1}{2b+1}}\log N$ where $b := \min(1, \alpha + \frac{1}{2})$ when Assumption 2 holds and $b = \frac{1}{2+c}$ when Assumption 1 holds instead.*

*Proof.* To control the error $\|G_{M,N} - G_W\|$ we decompose it into an estimation error $\|G_{M,N} - G_M\|$ and approximation error $\|G_M - G_W\|$:

$$\|G_{M,N} - G_W\| \le \|G_M - G_W\| + \|G_M - G_{M,N}\|$$

were $G_M$ is defined in (22) and is obtained by taking the number of samples $N$ to infinity while keeping the number of basis points $M$ fixed.

The estimation error $\|G_M - G_{M,N}\|$ is controlled using Proposition 17 where, for any $0 \le \delta \le 1$, we have with probability at least $1 - \delta$ and as long as $N \ge M(1, \lambda, \delta)$:

$$\|G_{M,N} - G_M\| \le \frac{\|B\|}{\sqrt{N\lambda}}(a_{N,\delta} + \sqrt{2\gamma_1 \kappa} + 2\gamma_1 \frac{\lambda + \kappa}{\sqrt{N\lambda}}) + \frac{1}{N\lambda}a_{N,\delta}^2.$$

In the limit where $N \to \infty$ and $\lambda \to 0$, only the dominant terms in the above equation remain which leads to an error $\|G_{M,N} - G_M\| = \mathcal{O}((N\lambda)^{-\frac{1}{2}})$. Moreover, the condition on $N$ can be expressed as $\lambda^{-1}\log\lambda^{-1} \lesssim N$.

To control the error approximation error $\|G_M - G_W\|$ we consider two cases: the *well-specified* case and the *miss-specified* case.

- *Well-specified* case. Here we work under Assumption 2 which allows to use Proposition 19. Hence, for any $0 \le \delta \le 1$ and if $M \ge M(d, \lambda, \delta)$, it holds with probability at least $1 - \delta$:

$$\|G_M - G_W\| \lesssim \lambda^{\min(1, \alpha + \frac{1}{2})}$$

- *Miss-specified* case. Here we work under Assumption 1 which allows to use Proposition 18. Hence, for any $0 \leq \delta \leq 1$ and if $M \geq M(d, \lambda, \delta)$, it holds with probability at least $1 - \delta$:

$$\|G_M - G_W\| \lesssim \lambda^{\frac{1}{2+c}}$$

Let's set $b := \min(1, \alpha + \frac{1}{2})$ for the well-specified case and $b = \frac{1}{2+c}$ for the miss-specified case. In the limit where $M \to \infty$ and $\lambda \to 0$ the condition on $M$ becomes: $M \sim d\lambda^{-1}\log\lambda^{-1}$. Hence, when $M \sim d\lambda^{-1}\log\lambda^{-1}$ and $\lambda^{-1}\log\lambda^{-1} \lesssim N$ it holds with probability as least $1 - \delta$ that

$$\|G_{M,N} - G_W\| = \mathcal{O}(\lambda^b + (\lambda N)^{-\frac{1}{2}}).$$

One can further choose $\lambda$ of the form $\lambda = N^{-\theta}$. This implies a condition on $M$ of the form $dN^\theta\log(N) \lesssim M$ and $N^\theta\log(N) \lesssim N$. After optimizing over $\theta$ to get the tightest bound, the optimal value is obtained when $\theta = 1/(2b+1)$ and the requirement on $N$ is always satisfied once $N$ is large enough. Moreover, one can choose $M \sim dN^{\frac{1}{2b+1}}\log N$ so that the requirement on $M$ is satisfied for $N$ large enough. In this case we get the following convergence rate:

$$\|G_{M,N} - G_W\| = \mathcal{O}(N^{-\frac{b}{2b+1}}).$$

$\square$

**Proposition 17.** *For any $0 \leq \delta \leq 1$, provided that $N \geq M(1, \lambda, \delta)$, we have with probability as least $1 - \delta$:*

$$\|G_{M,N} - G_M\| \leq \frac{\|B\|}{\sqrt{N\lambda}}(2a_{N,\delta} + \sqrt{2\gamma_1\kappa} + 2\gamma_1\frac{\lambda+\kappa}{\sqrt{N\lambda}}) + \frac{1}{N\lambda}a_{N,\delta}^2.$$

*with:*

$$a_{N,\delta} := \sqrt{2\sigma_1^2\log\frac{2}{\delta}} + \frac{2a\log\frac{2}{\delta}}{\sqrt{N}}$$

*Proof.* For simplicity, we define $E = \widehat{L^\top B} - L^\top B$. By definition of $G_{M,N}$ and $G_M$ we have the following decomposition:

$$G_{M,N} - G_M = \underbrace{E^\top\mathcal{G}_M(\hat{A}_\lambda)E}_{\mathfrak{E}_0} + \underbrace{E^\top\mathcal{G}_M(\hat{A}_\lambda)L^\top B}_{\mathfrak{E}_1} + \underbrace{B^\top L\mathcal{G}_M(\hat{A}_\lambda)E}_{\mathfrak{E}_2}$$
$$- \underbrace{B^\top L\mathcal{G}_M(A_\lambda)P_M(\hat{A}-A)P_M\mathcal{G}_M(\hat{A}_\lambda)L^\top B}_{\mathfrak{E}_3}$$

The first three terms can be upper-bounded in the following way:

$$\|\mathfrak{E}_0\| = \|E^\top\hat{A}_\lambda^{-\frac{1}{2}}\hat{A}_\lambda^{\frac{1}{2}}\mathcal{G}_M(\hat{A}_\lambda)\hat{A}_\lambda^{\frac{1}{2}}\hat{A}_\lambda^{-\frac{1}{2}}E\|$$
$$\leq \|E\|^2\underbrace{\|\hat{A}_\lambda^{-1}\|}_{\leq 1/\lambda}\underbrace{\|\hat{A}_\lambda^{\frac{1}{2}}\mathcal{G}_M(\hat{A}_\lambda)\hat{A}_\lambda^{\frac{1}{2}}\|}_{\leq 1}$$
$$\|\mathfrak{E}_1\| = \|\mathfrak{E}_2\| = \|E^\top A_\lambda^{-\frac{1}{2}}A_\lambda^{\frac{1}{2}}\mathcal{G}_M(A_\lambda)A_\lambda^{\frac{1}{2}}A_\lambda^{-\frac{1}{2}}L^\top B\|$$
$$\leq \|B\|\|E\|\underbrace{\|\hat{A}_\lambda^{-\frac{1}{2}}\|}_{\leq 1/\sqrt{\lambda}}\underbrace{\|\hat{A}_\lambda^{\frac{1}{2}}\mathcal{G}_M(\hat{A}_\lambda)\hat{A}_\lambda^{\frac{1}{2}}\|}_{\leq 1}\underbrace{\|A_\lambda^{-\frac{1}{2}}L^\top\|}_{\leq 1}\|\hat{A}_\lambda^{-\frac{1}{2}}A_\lambda^{\frac{1}{2}}\|$$

For the last term $\mathfrak{E}_3$ , we first recall that by definition of $\mathcal{G}_M(A_\lambda)$ we have:

$$\mathcal{G}_M(A_\lambda)P_M(\hat{A}-A)P_M\mathcal{G}_M(A_\lambda) = \mathcal{G}_M(A_\lambda)(\hat{A}-A)\mathcal{G}_M(A_\lambda).$$

Therefore, one can write:

$$\|\mathfrak{E}_3\| = \|B^\top LA_\lambda^{-\frac{1}{2}}A_\lambda^{\frac{1}{2}}\mathcal{G}_M(A_\lambda)A_\lambda^{\frac{1}{2}}A_\lambda^{-\frac{1}{2}}(\hat{A}-A)A_\lambda^{-\frac{1}{2}}A_\lambda^{\frac{1}{2}}\hat{A}_\lambda^{-\frac{1}{2}}\hat{A}_\lambda^{\frac{1}{2}}\mathcal{G}_M(\hat{A}_\lambda)\hat{A}_\lambda^{\frac{1}{2}}\hat{A}_\lambda^{-\frac{1}{2}}A_\lambda^{\frac{1}{2}}A_\lambda^{-\frac{1}{2}}L^\top B\|$$
$$\leq \|B\|^2\underbrace{\|LA_\lambda^{-\frac{1}{2}}\|^2}_{\leq 1}\underbrace{\|A_\lambda^{\frac{1}{2}}\mathcal{G}_M(A_\lambda)A_\lambda^{\frac{1}{2}}\|}_{\leq 1}\underbrace{\|\hat{A}_\lambda^{\frac{1}{2}}\mathcal{G}_M(\hat{A}_\lambda)\hat{A}_\lambda^{\frac{1}{2}}\|}_{\leq 1}\|A_\lambda^{\frac{1}{2}}\hat{A}_\lambda^{\frac{1}{2}}\|^2\|A_\lambda^{-\frac{1}{2}}(\hat{A}-A)A_\lambda^{-\frac{1}{2}}\|$$
$$\leq \|B\|^2\|A_\lambda^{\frac{1}{2}}\hat{A}_\lambda^{\frac{1}{2}}\|^2\|A_\lambda^{-\frac{1}{2}}(\hat{A}-A)A_\lambda^{-\frac{1}{2}}\|$$

We recall now (Rudi et al., 2015, Proposition 7.) which allows to upper-bound $\|A_\lambda^{\frac{1}{2}}\hat{A}_\lambda^{\frac{1}{2}}\|$ by $(1-\eta)^{-\frac{1}{2}}$ where $\eta = \lambda_{\max}(A_\lambda^{\frac{1}{2}}(A-\hat{A})A_\lambda^{\frac{1}{2}})$ provided that $\eta < 1$. Moreover, (Rudi et al., 2015, Proposition 8.) allows to control both $\eta$ and $\|A_\lambda^{-\frac{1}{2}}(\hat{A}-A)A_\lambda^{-\frac{1}{2}}\|$ under Assumption (F). Indeed, for any $0 \le \delta \le 1$ and provided that $0 < \lambda \le \|A\|$ it holds with probability $1-\delta$ that:

$$\|A_\lambda^{-\frac{1}{2}}(\hat{A}-A)A_\lambda^{-\frac{1}{2}}\| \le 2\gamma_1 \frac{1+\kappa/\lambda}{3N} + \sqrt{\frac{2\gamma_1\kappa}{N\lambda}}; \qquad \eta \le \frac{2\gamma_2}{3N} + \sqrt{\frac{2\gamma_2\kappa}{N\lambda}}$$

where $\gamma_1$ and $\gamma_2$ are given by:

$$\gamma_1 = \log(\frac{8Tr(A)}{\lambda\delta}); \qquad \gamma_2 = \log(\frac{4Tr(A)}{\lambda\delta}).$$

Hence, for $N \ge M(1,\lambda,\delta)$ we have that $(1-\eta)^{-\frac{1}{2}} \le 2$ and one can therefore write:

$$\|\mathfrak{E}_3\| \le 4\|B\|^2(2\gamma_1\frac{1+\kappa/\lambda}{3N} + \sqrt{\frac{2\gamma_1\kappa}{N\lambda}})$$

$$\|\mathfrak{E}_1\| = \|\mathfrak{E}_1\| \le \frac{2\|B\|}{\sqrt{\lambda}}\|E\|$$

The error $\|E\|$ is controlled by Proposition 22 where it holds with probability greater or equal to $1-\delta$ that:

$$\|E\| \le \frac{1}{\sqrt{N}}(\sqrt{2\sigma_1^2\log\frac{2}{\delta}} + \frac{2a\log\frac{2}{\delta}}{\sqrt{N}}) := \frac{1}{\sqrt{N}}a_{N,\delta}.$$

Finally, we have shown that provided that $N \ge M(1,\lambda,\delta)$ then with probability greater than $1-\delta$ one has:

$$\|G_{M,N} - G_M\| \le \frac{\|B\|}{\sqrt{N\lambda}}(2a_{N,\delta} + \sqrt{2\gamma_1\kappa} + 2\gamma_1\frac{\lambda+\kappa}{\sqrt{N\lambda}}) + \frac{1}{N\lambda}a_{N,\delta}^2.$$

$\square$

**Proposition 18.** *Let $0 \le \lambda \le \|A\|$ and define $M(d,\lambda,\delta) := \frac{128}{9}\log\frac{4Tr(A)}{\lambda\delta}(d\kappa\lambda^{-1}+1)$. Under Assumption 1 and Assumption (F), for any $\delta \ge 0$ such that $M \ge M(d,\lambda,\delta)$ the following holds with probability $1-\delta$:*

$$\|G_M - G_W\| \lesssim \lambda^{\frac{1}{2+c}}$$

*Proof.* We consider the error $\|G_M - G_W\|$. Recall that $G_W$ is given by $G_W = \Phi^\top\Phi$ with $\Phi$ defined in Proposition 13. Let $\kappa$ be a positive real number, we know by Assumption 1 that there exists $F^\kappa := (f_s^\kappa)_{1 \le s \le q}$ with $f_s^\kappa \in \mathcal{H}$ such that $\|\Phi - F^\kappa\|_{L_2(\rho_\theta)} \le C\kappa$ and $\|f_s^\kappa\|_{\mathcal{H}} \le C\kappa^{-c}$ for some fixed positive constant $C$. Therefore, we use $F^\kappa$ to control the error $\|G_M - G_W\|$. Let's call $E = \Phi \circ h_\theta - LF^\kappa$. We consider the following decomposition:

$$\begin{aligned} G_M - G_W &= (L^\top\Phi\circ h_\theta)^\top\mathcal{G}_M(A_\lambda)L^\top\Phi\circ h_\theta - \Phi^\top\Phi \\ &= \underbrace{E^\top L\mathcal{G}_M(A_\lambda)L^\top E}_{\mathfrak{E}_1} - \underbrace{E^\top E}_{\mathfrak{E}_2} \\ &\quad + \underbrace{F_\kappa^\top(L^\top L\mathcal{G}_M(A_\lambda)-I)L^\top\Phi\circ h_\theta}_{\mathfrak{E}_3} + \underbrace{E^\top L(\mathcal{G}_M(A_\lambda)L^\top L-I)F^\kappa}_{\mathfrak{E}_4} \end{aligned}$$

First we consider the term $\mathfrak{E}_1$ one simply has:

$$\|\mathfrak{E}_1\| \le \kappa^2 \underbrace{\|LA_\lambda^{-\frac{1}{2}}\|}_{\le 1}\underbrace{\|A_\lambda^{\frac{1}{2}}\mathcal{G}_M(A_\lambda)A_\lambda^{\frac{1}{2}}\|}_{\le 1}\underbrace{\|A_\lambda^{-\frac{1}{2}}L^\top\|}_{\le 1} \le \kappa^2$$

The second term also satisfies $\|\mathfrak{E}_1\| \le \kappa^2$ by definition of $F_\kappa$. For the last two terms $\mathfrak{E}_3$ and $\mathfrak{E}_4$ we use Lemma 20 which allows to control the operator norm of $L(\mathcal{G}_M(A_\lambda)L^\top L-I)$. Hence, for any $\delta \ge 0$ and $M$ such that $M \ge M(d,\lambda,\delta)$ and for $\kappa \le 1$ it holds with probability $1-\delta$ that:

$$\|\mathfrak{E}_3\| \lesssim \sqrt{\lambda}\kappa^{-c}; \qquad \|\mathfrak{E}_4\| \lesssim \sqrt{\lambda}\kappa^{-c}$$

We have shown so far that $\|G_M - G_W\| \lesssim (\kappa^2 + 2\kappa^{-c}\sqrt{\lambda})$. One can further optimize over $\kappa$ on the interval $[0,1]$ to get a tighter bound. The optimal value in this case is $\kappa^* = \min(1,(c\lambda^{\frac{1}{2}})^{\frac{1}{2+c}})$. By considering $\lambda > 0$ such that $(c\lambda^{\frac{1}{2}})^{\frac{1}{2+c}}) \le 1$, it follows directly that $\|G_M - G_W\| \lesssim \lambda^{\frac{1}{2+c}}$ which shows the desired result. $\square$

**Proposition 19.** *Let* $0 \le \lambda \le \|A\|$ *and define* $M(d,\lambda,\delta) := \frac{128}{9}\log\frac{4Tr(A)}{\lambda\delta}(d\kappa\lambda^{-1}+1)$. *Under Assumption 2 and Assumption (F), for any* $\delta \ge 0$ *such that* $M \ge M(d,\lambda,\delta)$ *the following holds with probability* $1-\delta$:

$$\|G_M - G_W\| \lesssim \lambda^{\min(1,\alpha+\frac{1}{2})}$$

*Proof.* Recall that $G_W$ is given by $G_W = \Phi^\top \Phi$ with $\Phi$ defined in Proposition 13. By Assumption 2, we have that $\Phi = \nabla(A^\alpha V)$ with $V := (v_s)_{1 \le s \le q} \in \mathcal{H}^q$. Hence, one can write

$$G_M - G_W = (L^\top \Phi \circ h_\theta)^\top \mathcal{G}_M(A_\lambda) L^\top \Phi \circ h_\theta - \Phi^\top \Phi$$
$$= V^\top (A^\alpha (A\mathcal{G}_M(A_\lambda)A - A)A^\alpha V$$

we can therefore directly apply Lemma 20 and get $\|G_M - G_W\| \lesssim \lambda^{\min(1,\alpha+\frac{1}{2})}$ with probability $1-\delta$ for any $\delta \ge 0$ such that $M \ge M(d,\lambda,\delta)$. $\qquad\square$

**Lemma 20.** *Let* $0 \le \lambda \le \|A\|$, $\alpha \ge 0$ *and define* $M(d,\lambda,\delta) := \frac{128}{9}\log\frac{4Tr(A)}{\lambda\delta}(d\kappa\lambda^{-1}+1)$. *Under Assumption (F), for any* $\delta \ge 0$ *such that* $M \ge M(d,\lambda,\delta)$ *the following holds with probability* $1-\delta$:

$$\|L(\mathcal{G}_M(A_\lambda)L^\top L - I)A^\alpha\| \lesssim \lambda^{min(1,\alpha+\frac{1}{2})}$$

*Proof.* We have the following identities:

$$L(\mathcal{G}_M(A_\lambda)L^\top L - I)A^\alpha = L(\mathcal{G}_M(A_\lambda)A_\lambda - I - \lambda\mathcal{G}_M(A_\lambda))A^\alpha$$
$$= \underbrace{LA_\lambda^{-\frac{1}{2}}A_\lambda^{\frac{1}{2}}(\mathcal{G}_M(A_\lambda)A_\lambda P_M - I)A^\alpha}_{\mathfrak{E}_1} - \underbrace{\lambda L A_\lambda^{-\frac{1}{2}}A_\lambda^{\frac{1}{2}}\mathcal{G}_M(A_\lambda)A_\lambda^{\frac{1}{2}}A_\lambda^{-\frac{1}{2}}A^\alpha}_{\mathfrak{E}_3}$$
$$+ \underbrace{LA_\lambda^{-\frac{1}{2}}A_\lambda^{\frac{1}{2}}\mathcal{G}_M(A_\lambda)A_\lambda^{\frac{1}{2}}A_\lambda^{\frac{1}{2}}(I-P_M)A^\alpha}_{\mathfrak{E}_2}.$$

For the first $\mathfrak{E}_1$ we use (Sutherland et al., 2017, Lemma 1 (iii)) which implies that $\mathcal{G}_M(A_\lambda)A_\lambda P_M = P_M$. Thus $\mathfrak{E}_1 = LA_\lambda^{-\frac{1}{2}}A_\lambda^{\frac{1}{2}}(P_M - I)A^\alpha$. Moreover, by Lemma 21 we have that $\|A_\lambda^{\frac{1}{2}}(I-P_M)\| \le 2\sqrt{\lambda}$ with probability $1 - \delta$ for $M > M(d,\lambda,\delta)$. Therefore, recalling that $(I - P_M)^2 = I - P_M$ since $P_M$ is a projection, one can further write:

$$\|\mathfrak{E}_1\| \le \underbrace{\|LA_\lambda^{-\frac{1}{2}}\|}_{\le 1}\underbrace{\|A_\lambda^{\frac{1}{2}}(P_M - I)\|^2}_{\le\lambda}\|A_\lambda^{-\frac{1}{2}}A^\alpha\|$$
$$\|\mathfrak{E}_2\| \le \underbrace{\|LA_\lambda^{-\frac{1}{2}}\|}_{\le 1}\underbrace{\|A_\lambda^{\frac{1}{2}}\mathcal{G}_M(A_\lambda)A_\lambda^{\frac{1}{2}}\|}_{\le 1}\underbrace{\|A_\lambda^{\frac{1}{2}}(P_M - I)\|^2}_{\le 4\lambda}\|A_\lambda^{-\frac{1}{2}}A^\alpha\|$$
$$\|\mathfrak{E}_3\| \le \lambda\underbrace{\|LA_\lambda^{-\frac{1}{2}}\|}_{\le 1}\underbrace{\|A_\lambda^{\frac{1}{2}}\mathcal{G}_M(A_\lambda)A_\lambda^{\frac{1}{2}}\|}_{\le 1}\|A_\lambda^{-\frac{1}{2}}A^\alpha\|$$

It remains to note that $\|A_\lambda^{-\frac{1}{2}}A^\alpha\| \le \lambda^{\alpha-\frac{1}{2}}$ when $0 \le \alpha \le \frac{1}{2}$ and that $\|A_\lambda^{-\frac{1}{2}}A^\alpha\| \le \|A\|^{\alpha-\frac{1}{2}}$ for $\alpha > \frac{1}{2}$ which allows to conclude. $\qquad\square$

## C.4 AUXILIARY RESULTS

**Lemma 21.** *Let* $0 \le \lambda \le \|A\|$. *Under Assumption (F), for any* $\delta \ge 0$ *such that* $M \ge M(d,\lambda,\delta) := \frac{128}{9}\log\frac{4Tr(A)}{\lambda\delta}(\kappa\lambda^{-1}+1)$ *the following holds with probability* $1-\delta$:

$$\|A_\lambda^{\frac{1}{2}}(I - P_M)\| \le 2\sqrt{\lambda}$$

*Proof.* The proof is an adaptation of the results in Rudi et al. (2015); Sutherland et al. (2017). Here we recall $Q_M$ defined in (21). Its transpose $Q_M^\top$ sends vectors in $\mathbb{R}^M$ to elements in the span of the Nyström basis

points, hence $P_M$ and $Q_M^\top$ have the same range, i.e.: $range(P_M) = \overline{range}(Q_M^\top)$. We are in position to apply (Rudi et al., 2015, Proposition 3.) which allows to find an upper-bound on $A_\lambda^{\frac{1}{2}}(P_M - I)$ in terms of $Q_M$:

$$\|A_\lambda^{\frac{1}{2}}(P_M - I)\| \leq \sqrt{\lambda}\|A_\lambda^{\frac{1}{2}}(Q_M^\top Q_M + \lambda I)^{-\frac{1}{2}}\|.$$

For simplicity we write $\hat{A}_M := Q_M^\top Q_M$ and $E_2 := A_\lambda^{-\frac{1}{2}}(A - \hat{A}_M)A_\lambda^{-\frac{1}{2}}$. We also denote by $\beta = \lambda_{max}(E_2)$ the highest eigenvalue of $E_2$. We can therefore control $\|A_\lambda^{\frac{1}{2}}(\hat{A}_M + \lambda I)^{-\frac{1}{2}}\|$ in terms of $\beta$ using (Rudi et al., 2015, Proposition 7) provided that $\beta < 1$:

$$\|A_\lambda^{\frac{1}{2}}(P_M - I)\| \leq \sqrt{\lambda}\frac{1}{\sqrt{1-\beta}}.$$

Now we need to make sure that $\beta < 1$ for $M$ large enough. To this end, we will apply (Rudi et al., 2015, Proposition 8.) to $\hat{A}_M$. Denote by $v_m = \sqrt{d}\partial_{i_m}k(Y_m,.)$. Hence, by definition of $\hat{A}_M$ it follows that $\hat{A}_M = \frac{1}{M}\sum_{m=1}^M v_m \otimes v_m$. Moreover, $(v_m)_{1 \leq m \leq M}$ are independent and identically distributed and satisfy:

$$\mathbb{E}[v_m \otimes v_m] = \int \sum_{i=1}^q \partial_i k(y,.) \otimes \partial_i k(y,.) \mathrm{d}p_\theta(y) = A.$$

We also have by Assumption (F) that $\langle v_m, A_\lambda^{-1} v_m \rangle \leq \frac{d\kappa}{\lambda}$ almost surely and for all $\lambda > 0$. We can therefore apply (Rudi et al., 2015, Proposition 8.) which implies that for any $1 \geq \delta \geq 0$ and with probability $1 - \delta$ it holds that:

$$\beta \leq \frac{2\gamma}{3M} + \sqrt{\frac{2\gamma d\kappa}{M\lambda}}$$

with $\gamma = \log\frac{4Tr(A)}{\lambda\delta}$ provided that $\lambda \leq \|A\|$. Thus by choosing $M \geq \frac{128\gamma}{9}(d\kappa\lambda^{-1} + 1)$ we have that $\beta \leq \frac{3}{4}$ with probability $1 - \delta$ which allows to conclude. $\square$

**Proposition 22.** *There exist $a > 0$ and $\sigma_1 > 0$ such that for any $0 \leq \delta \leq 1$, it holds with probability greater of equal than $1 - \delta$ that:*

$$\|\widehat{L^\top B} - L^\top B\| \leq \frac{2a\log\frac{2}{\delta}}{N} + \sqrt{\frac{2\sigma_1^2\log\frac{2}{\delta}}{N}}$$

*Proof.* denote by $v_n = \sum_{i=1}^d \partial_i k(X_n,.)B_i(Z_n)$, we have that $\mathbb{E}[v_n] = L^\top B$. We will apply Bernstein's inequality for sum of random vectors. For this we first need to find $a > 0$ and $\sigma_1 > 0$ such that $\mathbb{E}[\|z_n - L^\top B\|_\mathcal{H}^p] \leq \frac{1}{2}p!\sigma_1^2 a^{p-2}$. To simplify notations, we write $x$ and $x'$ instead of $h_\theta(z)$ and $h_\theta(z')$. We have that:

$$\mathbb{E}[\|z_n - L^\top B\|_\mathcal{H}^p] = \int \left\|\sum_{i=1}^d \partial_i k(x,.)B_i(z) - \int \sum_{i=1}^d \partial_i k(x',.)B_i(z')\mathrm{d}\nu(z')\right\|^p \mathrm{d}\nu(z)$$

$$\leq 2^{p-1}\underbrace{\int\left\|\sum_{i=1}^d \int(\partial_i k(x,.) - \partial_i k(x',.))B_i(z)\mathrm{d}\nu(z')\right\|^p \mathrm{d}\nu(z)}_{\mathfrak{E}_1}$$

$$+ 2^{p-1}\underbrace{\int\left\|\int\sum_{i=1}^d \partial_i k(x,.)(B_i(z) - B_i(z'))\mathrm{d}\nu(z')\right\|^p \mathrm{d}\nu(z)}_{\mathfrak{E}_2}$$

We used the convexity of the norm and the triangular inequality to get the last line. We introduce the notation $\gamma_i(x) := \partial_i k(x,.) - \int \partial_i k(h_\theta(z'),.)\mathrm{d}\nu(z')$ and by $\Gamma(x)$ we denote the matrix whose components are given by $\Gamma(x)_{ij} := \langle\gamma_i(x), \gamma_j(x)\rangle_\mathcal{H}$. The first term $\mathfrak{E}_1$ can be upper-bounded as follows:

$$\mathfrak{E}_1 = \int\left|Tr(B(z)B(z)^\top\Gamma(x))\right|^{\frac{p}{2}}$$

$$\leq \int\left|\|B(z)\|^2 Tr(\Gamma(x)^2)^{\frac{1}{2}}\right|^{\frac{p}{2}}.$$

Moreover, we have that $Tr(\Gamma(x)^2)^{\frac{1}{2}} = (\sum_{1 \le i,j \le d} \langle \gamma_i(x), \gamma_j(x) \rangle_{\mathcal{H}}^2)^{\frac{1}{2}} \le \sum_{i=1}^d \|\gamma_i(x)\|^2$. We further have that $\|\gamma_i(x)\| \le \partial_i \partial_{i+d} k(x,x)^{\frac{1}{2}} + \int \partial_i \partial_{i+d} k(h_\theta(z), h_\theta(z))^{\frac{1}{2}} \mathrm{d}\nu(z)$ and by Assumption (F) it follows that $\|\gamma_i(x)\| \le 2\sqrt{\kappa}$. Hence, one can directly write that: $\mathfrak{E}_1 \le (2\sqrt{\kappa d})^p \int \|B(z)\|^p \mathrm{d}\nu(z)$. Recalling Assumptions (B) and (C) we get:

$$\mathfrak{E}_1 \le 2^{p-1}(2\sqrt{\kappa d})^p C(\theta)^p (1 + \frac{1}{2}p! \zeta^{p-2} \sigma^2)$$

Similarly, we will find an upper-bound on $\mathfrak{E}_2$. To this end, we introduce the matrix $Q(x', x'')$ whose components are given by $Q(x', x'')_{i,j} = \partial_i \partial_{i+d} k(x', x'')$. One, therefore has:

$$\mathfrak{E}_2 = \int \left| \int \int Tr((B(z) - B(z'))(B(z) - B(z''))^\top Q(x', x'') \mathrm{d}\nu(z') \mathrm{d}\nu(z'') \right|^{\frac{p}{2}} \mathrm{d}\nu(z)$$

$$\le \int \left| \int \int \|B(z) - B(z')\| \|B(z) - B(z'')\| Tr(Q(x', x'')^2)^{\frac{1}{2}} \mathrm{d}\nu(z') \mathrm{d}\nu(z'') \right|^{\frac{p}{2}} \mathrm{d}\nu(z)$$

Once again, we have that $Tr(Q(x', x'')^2)^{\frac{1}{2}} \le (\sum_{i=1}^d \partial_i \partial_{i+d} k(x', x'))^{\frac{1}{2}} (\sum_{i=1}^d \partial_i \partial_{i+d} k(x'', x''))^{\frac{1}{2}} \le d\kappa$ thanks to Assumption (F). Therefore, it follows that:

$$\mathfrak{E}_2 \le (\sqrt{d\kappa})^p \int |\int \|B(z) - B(z')\| \mathrm{d}\nu(z)|^p \mathrm{d}\nu(z)$$

$$\le 3^{p-1}(\sqrt{d\kappa})^p C(\theta)^p (2^p + \int \|z\|^p \mathrm{d}\nu(z) + \left(\int \|z\| \mathrm{d}\nu(z)\right)^p)$$

$$\le 3^{p-1}(\sqrt{d\kappa})^p C(\theta)^p (2^p + \frac{1}{2}p! \zeta^{p-2} \sigma^2 + \left(\int \|z\| \mathrm{d}\nu(z)\right)^p).$$

The second line is a consequence of Assumption (C) while the last line is due to Assumption (B). These calculations, show that it is possible to find constants $a$ and $\sigma_1$ such that $\mathbb{E}[\|z_n - L^\top B\|_{\mathcal{H}}^p] \le \frac{1}{2}p! \sigma_1^2 a^{p-2}$. Hence one concludes using Bernstein's inequality for a sum of random vectors (see for instance Rudi et al., 2015, Proposition 11). $\qquad \square$

# D EXPERIMENTAL DETAILS

## D.1 NATURAL WASSERSTEIN GRADIENT FOR THE MULTIVARIATE NORMAL MODEL

**Multivariate Gaussian.** Consider a multivariate gaussian with mean $\mu \in \mathbb{R}^d$ and covariance matrix $\Sigma \in \mathbb{R}^d \times \mathbb{R}^d$ parametrized using its lower triangular components $s = T(\Sigma)$. We denote by $\Sigma = T^{-1}(s)$ the inverse operation that maps any vector $s \in \mathbb{R}^{\frac{d(d+1)}{2}}$ to its corresponding symmetric matrix in $\mathbb{R}^d \times \mathbb{R}^d$. The concatenation of the mean $\mu$ and $s$ will be denoted as $\theta : \theta = (\mu, s)$. Given two parameter vectors $u = (m, T(S))$ and $v = (m', T(S'))$ where $m$ and $m'$ are vectors in $\mathbb{R}^d$ and $S$ and $S'$ are symmetric matrices in $\mathbb{R}^d \times \mathbb{R}^d$ the metric evaluated at $u$ and $v$ is given by:

$$u^\top G(\theta) v = m^\top m' + Tr(A\Sigma A')$$

where $A$ and $A'$ are symmetric matrices that are solutions to the Lyapunov equation:

$$S = A\Sigma + \Sigma A, \qquad S' = A'\Sigma + \Sigma A'.$$

$A$ and $A'$ can be computed in closed form using standard routines making the evaluation of the metric easy to perform. Given a loss function $\mathcal{L}(\theta)$ and gradient direction $\nabla_\theta \mathcal{L}(\theta) = \nabla_\mu \mathcal{L}(\theta), \nabla_s \mathcal{L}(\theta)$, the corresponding natural gradient $\nabla_\theta^W \mathcal{L}(\theta)$ can also be computed in closed form:

$$\nabla_\theta^W \mathcal{L}(\theta) = (\nabla_\mu \mathcal{L}(\theta), T(\Sigma(A + diag(A)) + (A + diag(A))\Sigma)),$$

where $A = T^{-1}(\nabla_s \mathcal{L}(\theta))$. To use the estimator proposed in Proposition 5 we take advantage of the parametrization of the Gaussian distribution as a push-forward of a standard normal vector:

$$X \sim \mathcal{N}(\mu, \Sigma) \iff X = \Sigma^{\frac{1}{2}} Z + \mu, Z \sim \mathcal{N}(0, I_d)$$

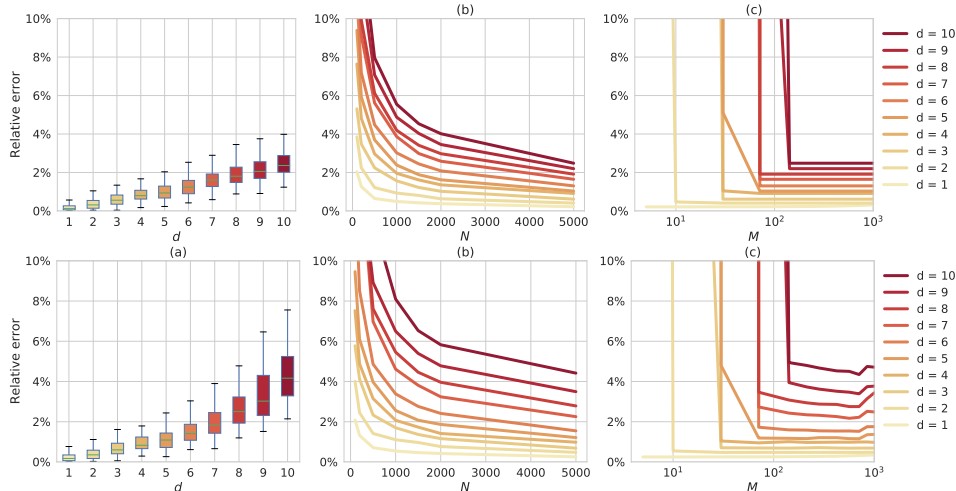

Figure 4: Evolution of the relative error of KWNG averaged over 100 runs for varying dimension form $d=1$ (yellow) to $d=10$ (dark red). For each run, a random value for the parameter $\theta$ and for the Euclidean gradient $\nabla\mathcal{L}(\theta)$ is sampled from a centered Gaussian with variance 0.1. In all cases, $\lambda=0$ and $\epsilon=10^{-5}$. Top row: multivariate normal model, bottom row: multivariate log-normal. Left (a): box-plot of the relative error as $d$ increases with $N=5000$ and the number of basis points is set to $M=\left\lfloor d\sqrt{N}\right\rfloor$. (b) Relative error as the sample size $N$ increases and the number of basis points is set to $M=\left\lfloor d\sqrt{N}\right\rfloor$. Right (c): Relative error as $M$ increases and $N$ fixed to 5000.

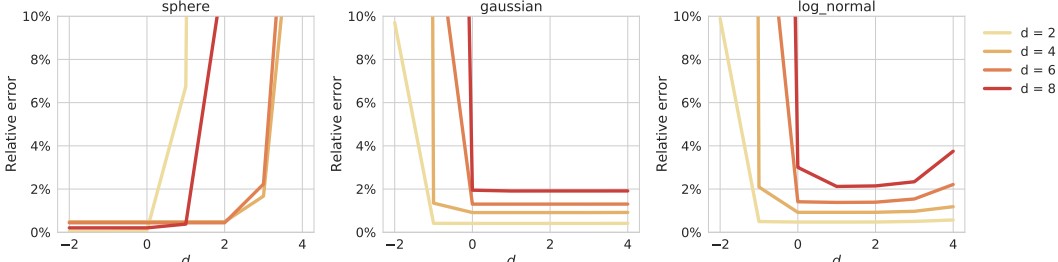

Figure 5: Relative error of the KWNG for varying bandwidth of the kernel. Results are averaged over 100 runs for varying dimension form $d=1$ (yellow) to $d=10$ (dark red). For each run, a random value for the parameter $\theta$ and for the Euclidean gradient $\nabla\mathcal{L}(\theta)$ is sampled from a centered Gaussian with variance 0.1. In all cases, $\lambda=\epsilon=10^{-10}$. The sample size is fixed to $N=5000$ and the number of basis points is set to $M=\left\lfloor d\sqrt{N}\right\rfloor$. Left: uniform distributions on a hyper-sphere, middle: multivariate normal, and right: multivariate log-normal.

### D.2  CLASSIFICATION ON CIFAR10 AND CIFAR100

**Architecture.**  We use a residual network with one convolutional layer followed by 8 residual blocks and a final fully connected layer. Each residual block consists of two $3\times3$ convolutional layers each and ReLU nonlinearity. We use batch normalization for all methods. Details of the intermediate output shapes and kernel size are provided in Table 1.

**Hyper-parameters.**  For all methods, we used a batch-size of 128. The optimal step-size $\gamma$ was selected in $\{10,1,10^{-1},10^{-2},10^{-3},10^{-4}\}$ for each method. In the case of SGD with momentum, we used a momentum parameter of 0.9 and a weight decay of either 0 or $5\times10^{-4}$. For KFAC and EKFAC, we used a damping coefficient of $10^{-3}$ and a frequency of reparametrization of 100 updates. For KWGN we set $M=5$ and $\lambda=0$ while the initial value for $\epsilon$ is set to $\epsilon=10^{-5}$ and is adjusted using an adaptive scheme based on the Levenberg-Marquardt dynamics as in (Martens and Grosse, 2015, Section 6.5). More

| | Kernel size | Output shape |
|---|---|---|
| z | | $32 \times 32 \times 3$ |
| Conv | $3 \times 3$ | 64 |
| Residual block | $[3 \times 3] \times 2$ | 64 |
| Residual block | $[3 \times 3] \times 2$ | 128 |
| Residual block | $[3 \times 3] \times 2$ | 256 |
| Residual block | $[3 \times 3] \times 2$ | 512 |
| Linear | - | Number of classes |

Table 1: Network architecture.

precisely, we use the following update equation for $\epsilon$ after every 5 iterations of the optimizer:

$$\epsilon \leftarrow \omega\epsilon, \qquad\qquad\qquad \text{if } r > \frac{3}{4}$$

$$\epsilon \leftarrow \omega^{-1}\epsilon, \qquad\qquad\qquad \text{if } r < \frac{1}{4}.$$

Here, $r$ is the reduction ratio:

$$r = \max_{t_{k-1} \leq t \leq t_k} \left( 2\frac{\mathcal{L}(\theta_t)) - \mathcal{L}(\theta_{t+1})}{\nabla^W \mathcal{L}(\theta)^\top \nabla \mathcal{L}(\theta)^\top} \right)$$

where $(t_k)_k$ are the times when the updates occur. and $\omega$ is the decay constant chosen to $\omega = 0.85$.

## D.3 ADDITIONAL EXPERIMENTS

Figure 6 reports the time cost in seconds for each methods, while Figure 7 compares KWNG to diagonal conditioning.

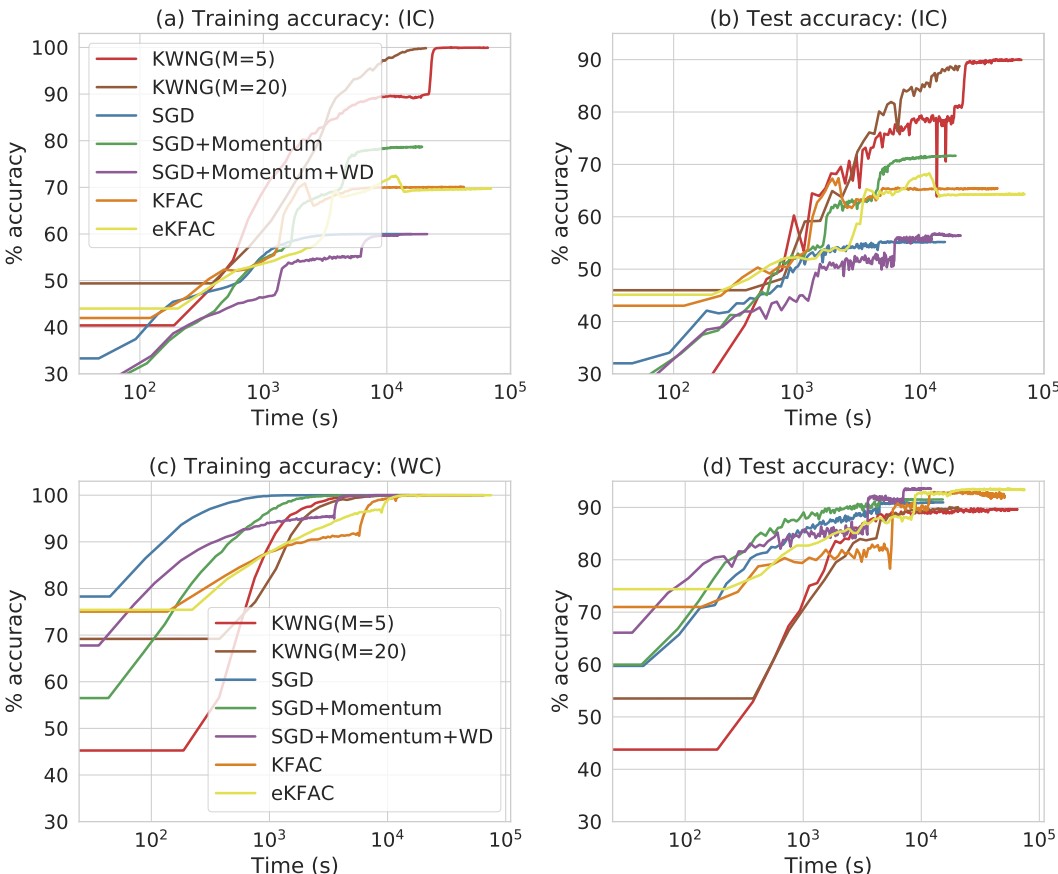

Figure 6: Training accuracy (left) and test accuracy (right) as a function of time for classification on `Cifar10` in both the ill-conditioned case (top) and well-conditioned case (bottom) for different optimization methods.

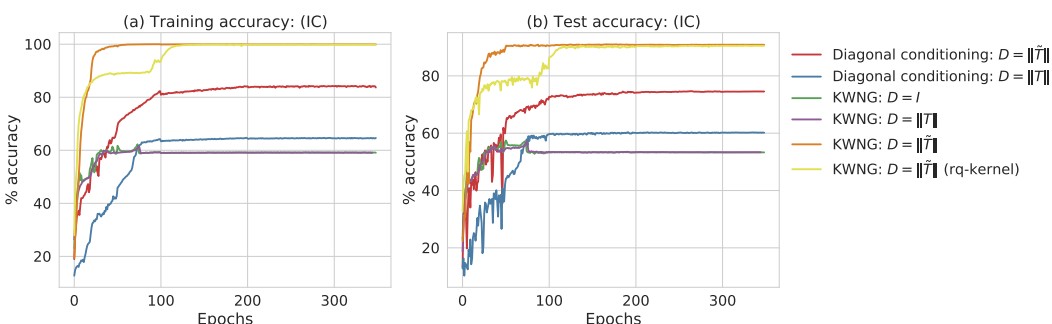

Figure 7: KWNG vs Diagonal conditioning in the ill-conditioned case on Cifar10. In red and blue, the euclidean gradient is preconditioned using a diagonal matrix $D$ either given by $D_i = \|T_{.,i}\|$ or $D_i = \|\widetilde{T}_{.,i}\|$, where $T$ and $\widetilde{T}$ are defined in Propositions 5 and 6. The rest of the traces are obtained using the stable version of KWNG in Proposition 6 with different choices for the damping term $D = I$ , $D = \|T_{.,i}\|$ and $\|\widetilde{T}_{.,i}\|$. All use a gaussian kernel except the yellow traces which uses a rational quadratic kernel.

