# OpenReview forum: "Kernelized Wasserstein Natural Gradient"
_ICLR.cc/2020/Conference — Accept (Spotlight)_

### Official Review · AnonReviewer3 · 2019-10-22
**Official Blind Review #3**

**Rating:** 6

**Review:**

The authors propose an approximate of the natural gradient under Wasserstein metric when optimizing some cost function over a parametric family of probability distributions. The authors leverage the dual formulation and restrict the feasible space to a RKHS. The authors show a trade-off between accuracy and computational cost with theoretical guarantees for the proposed method, and empirically verify it for classification tasks.

The motivation of the natural gradient is well-motivated. Although the choice of Wasserstein metric is sound, especially for models that do not admit a density, it seems that there are no supporting experiments for this choice over Fisher Information metric. In general, the writing is fine. The flow idea is clear. However, the content is quite dense. Some assumptions just pump out without careful judgment. The idea to restrict into RKHS and use low-rank approach is interesting to approximate for the natural gradient under Wasserstein metric. Overall, I lean to the acceptance side.

Below are some of my concerns:

1) It seems that the natural gradient under Wasserstein metric is well-motivated for models which do not admit a density (to compare with the natural gradient under Fisher information metric). However, it seems that there is no supporting experiments about it yet. For models in the experiments, it is better to show a comparison between natural gradient under Wasserstein metric and Fisher information metric w.r.t. time consumption and accuracy.

2) In proposition 3 and theorem 5, they require some assumptions. The authors should place those assumptions into the main text instead of only putting it in the appendix, and should give more discussions about those assumptions. Especially, for assumption (D), why one can have this assumption? It seems that this assumption (D) has a strong influence to the complexity in Theorem 5? More detail discussion is required.

3) For the relaxation in Equation (9), it seems that the authors do not simply add some regularization terms. How does it relate to the original Equation (7)? What is the meaning of the 3rd term in Equation (9)? and how's about the 2nd term?

4) For the experiments, the authors evaluate the multivariate normal model and the multivariate log-normal model which are very special cases under Wasseserstein information matrix where one can compute in closed-form. The authors should show some general models, especially models which do not admit a density. For the experiments in Section 4.2, the authors should add the natural baseline: natural gradient under Fisher information metric. It is unclear to me why one needs natural gradient under Wasserstein metric over Fisher information metric for this setup? What is the benefit to use natural gradient under Wasserstein metric?



**Experience Assessment:**

I have read many papers in this area.

**Review Assessment: Checking Correctness Of Derivations And Theory:**

I assessed the sensibility of the derivations and theory.

**Review Assessment: Checking Correctness Of Experiments:**

I assessed the sensibility of the experiments.

**Review Assessment: Thoroughness In Paper Reading:**

I read the paper at least twice and used my best judgement in assessing the paper.

---

> ### Author Response · Authors · 2019-11-15
> **Reply**
>
> Thanks for your comments. In the revision we've posted, we addressed most of them. Here is a summary:
>
> 1- Models with no density: we replaced the gaussian model by a model consisting of hyper-spheres in R^d parametrized by their radii and centers. These do not admit a density in R^d and do not share the same support, thus the Fisher natural gradient is not defined.
>
> 2- Comparison with the Fisher gradient:  The KFAC and eKFAC method estimate the fisher natural gradient, and we compare the proposed method with those in terms of accuracy Figure 3 and timing Figure 6. That being said, we have  not yet implemented the  estimator for the Fisher gradient suggested by the variational formulation of Proposition 2: this is an interesting topic for future work.
>
> 3- Assumptions: we state the assumptions briefly in the main text and provide a discussion for when assumption (D) holds:   It is a mild assumption that is satisfied for instance in the case where the gradient is an empirical mean of iid samples with finite variance. This result from Chebychev’s inequality: we mention this in the main text and further discuss it in Remark 1 of appendix A.2.
> The remaining assumptions are also mild, in  that the kernel can be chosen to satisfy them while the assumption on the implicit model is often satisfied in many cases: especially in the case of deep networks with ReLU non-linearity.
>
> 4- Relaxation of equation in (9)  ( (11) in the revised version). We make the connection more explicit by first provide the full variational expression for the natural gradient in Prop3, so that the difference is simply restricting the optimization and adding the two penalization terms.We also include the following comment to clarify the purpose of the regularization terms:
> “The first regularization term makes the problem strongly convex in u, while the second term makes the problem strongly concave in f when \lambda  > 0. When \lambda=0, the problem is still concave in f. This allows to use a version of the minimax theorem (Ekeland and Temam,1999)” to exchange the order of the supremum and minimum which also holds true when \lambda=0.
> The result of prop 3 (prop 4 in the revised version) holds without further assumptions since the problem is strongly convex in u and concave in f.
>
> 5- Wasserstein vs Fisher:  both allow to get optimization trajectories that are invariant to parametrization, as discussed in Prop 1. From this point of view, there is no reason to prefer one over the other. The only difference is that the Wasserstein can be used in settings where the Fisher cannot be used. This is the case in the  hyper-sphere model, for which the Wasserstein natural gradient can be estimated accurately, as illustrated in figure 1.
>
> 6- A possible use case of the wasserstein would be for learning the policy of an actor- critic in reinforcement learning. In [Schulman et al. 2015] it was shown that using the Fisher natural gradient can improve the training of a policy when its density is available. Recently, [Tang and Agrawal 2019] considered a new class of policies that are parametrized implicitly. These new category of policies were motivated by the success of implicit generative models. In this case, however, the density is not available explicitly and might not be well defined. Using the Wasserstein natural gradient could lead to similar improvements as observed for in  [Schulman et al. 2015] for the case when the Fisher natural gradient can be used. We leave this as a future research direction.

---

### Official Review · AnonReviewer2 · 2019-10-23
**Official Blind Review #2**

**Rating:** 8

**Review:**

Thank you for your revision and for the rebuttal. This a  strong submission with insightful angle on natural gradients and with provable guarantees. The authors improved a lot the manuscript and incorporated reviewers feedback. I am increasing my score to 8.

###
Summary of the paper:

The paper provides a way to estimate the natural Wasserstein gradient using Kernel estimators. The idea is neat and novel. Natural Wasserstein Gradient similar to the so called natural fisher gradients preconditions the gradient using a matrix that uses the local curvature of the manifold of the parametric distribution.

Authors give variational forms of the Fisher information matrix of an explicit model , using  the variational form of the chi squared or the Fisher Rao divergence. Similarly authors give a variational form of the wasserstein natural gradient . Let theta be the parameter of the parametric implicit model, theta in R^q.  For a descent direction $u$, the variational form is obtained via finding an objective S,  $\sup_{f\in C^c_{\infty}} S(f, u ) = u^{\top}G_{W}u$, where $G_{W}$ is a form of  "Wasserstein information matrix".

Authors then propose to learn the function f in an RKHS and propose to find the descent direction by solving
 $\min_{u} <u, \nabla_{\theta} Loss (p_{\theta})>  + \sup_{f\in RKHS}  S(f, u ) + r(u)- \lambda ||f||^2_{rkhs}$
where r(u) is a quadratic regularizer on u.

The sup problem has a closed form solution and can be approximated  using Nystrom approximation and randomization on dimensions. The problem in u has also a closed form solution , and one used u as the proxy to the natural  descent.

Authors under some assumption  show that the  estimated natural W gradient in RKHS  is concentrated around the true one.

Experiments on synthetic data and in classification on CIFAR 10 and CIFAR 100 shows that the preconditioning of the gradients that the method offers allows faster convergence in both well conditioned and ill conditioned initialization of the weights of the neural network.

Review :

The paper is not easy to follow and the high level intuition how the method works is not well explained.

It would be easier for the reader, to motivate the natural wasserstein descent from how one defines natural Fisher descent , where one seeks a first order approximation of $KL(p_{\theta},p_{\theta+ \epsilon u})$ as we perturb in the parameter space and this well known that this epsilon $u^{\top}F u$.
Hence natural Gradient descent is :
$\min _{u} <u, \nabla_{\theta}Loss(p_{\theta})> + KL(p_{\theta},p_{\theta+  u}) \approx min _{u} <u, \nabla_{\theta}Loss(p_{\theta})> + u^{\top}F u$

Now for the wasserstein distance one has also similarly:
$\min _{u} <u, \nabla_{\theta}Loss(p_{\theta})> + W^2_2(p_{\theta},p_{\theta+  u})$

and it is known that as epsilon goes to zero we have:

 $W^2_2(p_{\theta},p_{\theta+ \epsilon u})/\epsilon = ||p_{\theta} - p_{\theta+ \epsilon u}||^2_{H^{-1}(p_{\theta})}+o(\epsilon) = \sup_{f} \int f (p_{\theta} - p_{\theta+\epsilon u}) - \frac{1}{2} \mathbb{E}_{p_{\theta}}||\nabla_x f(x)||^2+o(\epsilon) $

Now replacing with the implicit model as epsilon goes to zero we get the expression given in the paper using a simple taylor expansion:
$=   \sup_{f} \int <\nabla_{\theta} h_{\theta}^{\top}\nabla_x f(h_{\theta}),u > d\nu - \frac{1}{2}\mathbb{E} _{p_{\theta}}||\nabla_x f(x)||^2$

in a sense the paper is proposing to linearize $W^2_2$ around the perturbation in the parameter space of the implicit model and this can be done using  $|| .||_{H^{-1}(q)}$ , as pointed and used in many recent works.  then the paper proposes to approximate $|| .||_{H^{-1}(q)}$ in RKHS which was already proposed in Mroueh et al  in Sobolev Descent.  AISTATS 2019.

We encourage the authors to layout in the beginning the derivations form this point of view which will make the paper easier to digest, the expression in Equation 7 seems mysterious and pulled out of a hat, but it is easier to understand by going to perturbation analysis usually done on KL for Fisher Natural gradient and to do it also here starting from the linearization of $W_2$ with $||.||_{H^{-1}(q)}$  , and how to approximate it in RKHS as it was already proposed in the literature in Mroueh et al Sobolev Descent.


I read carefully the proofs of Proposition 1, 2, 3. I did not ready full the proofs of the concentration of the estimator , but they seem sensible as they follow usual bounding strategies in this context.

Questions:

- There is nothing special about the wasserstein natural gradient flow variational form and implicit model, once can apply the same to the variational form of Fisher, that would be probably more efficient? It would be great to baseline this one ?
-the constraint $\int f(x)p_{\theta}(x)=0$ is not imposed in the kernelized version?
- the method comes disappointing since it seems that the preconditioning that the Wasserstein gradient gives is not enough and $r(u)=u^{\top}D u$ is need where D is diagonal depends on T. Have you tried with $D=Identity$? it might be that the scaling of the gradients is coming only from that $D^{-1}$?

- Can you give timings for computing each gradient update and how it compares to regular SGD or diagonal approximation of Fisher natural gradient?

- Does one need preconditioned gradient if the network was self normalized (like batch norm or spectral norm etc)?


Overall assessment:

That is a good theoretical work with provable guarantees. The computational complexity of each gradient estimate is large which makes the method not quite appealing in practice.

**Experience Assessment:**

I have published in this field for several years.

**Review Assessment: Checking Correctness Of Derivations And Theory:**

I assessed the sensibility of the derivations and theory.

**Review Assessment: Checking Correctness Of Experiments:**

I carefully checked the experiments.

**Review Assessment: Thoroughness In Paper Reading:**

I read the paper thoroughly.

---

> ### Author Response · Authors · 2019-11-15
> **Reply**
>
> Thank you for the useful feedback about the high level intuition and the connection with the work in AISTATS 2019.
>
> 1- We adapted section 2.1 to first present the general perturbative approach for any divergence of distance, we then discuss the particular case of the wasserstein in more detail in Appendix B.3 as well as the connection with the Negative sobolev distance. We highlight the fact that a kernel method was proposed for the negative sobolev distance in the introduction and at the end of section 2.3 and further comment on it after equation 14.
> in Appendix B.3, we also discuss the difference between the Negative Sobolev distance and the wasserstein metric. In particular, we show that for models with disjoint support -  namely dirac distributions - the negative  sobolev distance is always infinite while the wasserstein metric will still be finite. We conclude that under some additional regularity assumptions, the wasserstein metric can be obtained as a limit of the Negative Sobolev distance.
>
> 2- Fisher gradient: Indeed a similar variational form for the fisher gradient holds as made more explicit in Proposition 2 of the revised version. One can also use an implicit model in the case of the Fisher gradient. However, if the model doesn’t admit a density, the variational expression will be infinite. As an illustration, this can be done in the case of the normal and log-normal distributions considered in Figure 4, but not in the case of the hyper-sphere (figure 1). We did not yet implement the variational form for the Fisher, however - this will be an interesting topic for future work. On the other hand, KFAC and eKFAC implement an approximation to the Fisher gradient to which we compare our method on cifar10 an cifar100.
>
> 3- The constraint:  we corrected this in the revised version. The constraint is not needed in the case of the wasserstein distance because the objective function appearing in the variational formulation only depends on derivatives of f.
>
> 4 -Diagonal conditioning:  We run an ablation study to analyse the contribution of the diagonal scaling alone. Figure 7 (red) shows that using the same diagonal conditioning D instead of KWNG doesn’t match the performance of the proposed method:  (76% test accuracy vs 90% for KWNG ). However, it slightly improves on to second best method which was  SGD with Momentum (73% test accuracy) Figure 3.
> When setting the Diagonal term to identity, KWNG is not as effective and its performance drops to 60% which is the same performance as plain SGD (Figure 3). This is consistent with the discussion in section 3.3 about the choice of damping which is particularly important in the ill-conditioned case where the hessian is far from being isotropic. We also note that this behavior is not specific to KWNG, and affects many stochastic second order methods, as discussed in [Martens and Sutskever (section 8.2)].
>
> 5- Batch-normalization: all experiments used batch-normalization. While we found it helpful in general, it didn’t overcome the ill-conditioning case, so the preconditioning was still required.
> While spectral normalization ensures the highest eigenvalues are less than one, in principle it doesn’t affect the conditioning of the weights. However,  we haven’t tried it in this work and leave it for future work.
>
> 6- Timing: Using the same setting:  batch-normalization and gradient clipping by norm, the cost per iteration is almost 4.7 times larger, which corresponds approximately to the cost of performing the additional 5 backward passes required by the algorithm to compute the nystrom approximation of the natural gradient:  0.61s vs 0.13s. Figure 6 (appendix D.1) compares the evolution of the error per second. It is favorable for KWNG in the case of ill-conditioned models and is comparable with the other methods.

---

### Official Review · AnonReviewer1 · 2019-10-24
**Official Blind Review #1**

**Rating:** 6

**Review:**

Natural gradient has been proven effective in many statistical learning algorithms. A well-known difficulty in using natural gradient is that it is tedious to compute the Fisher matrix (if one is using Fisher-Rao metric) and the Wasserstein information matrix (if one is using Wasserstein metric). It's important to be able to estimate natural gradient in a practical way, and there have been a few papers looking at this problem but mostly for the case with a Fisher-Rao metric. This paper takes a different and general approach to approximate natural gradient by leveraging the dual formulation for the metric restricted to the Reproducing Kernel Hilbert Space. Some theoretical guarantees of the proposed method is established together to some experimental study.

I find this work interesting with some important merit, as it tackles an important problem in statistical learning. My main concern, however, is the problems related to RKHS from a practical point of view. For example, solving optimization problem (11) is difficult and the paper makes a range of further approximations to be able to arrive at an approximate solution. Also, selecting the kernel and its bandwidth is crucial in practice. From a practical point of view, I suspect that more evidence is needed to justify if the proposed method can really offer a method of choice.

Having said that, I believe this paper provides an important first (and alternative) step towards an important problem. The paper is also well written and well structured. I have a few further comments below
1) In the abstract and introduction, the invariant property of natural gradient is mentioned several times without a detailed explanation why/what it is. Adding a brief explanation of this property is appreciated.
2) The sentence on line 8 in Introduction reads ".. It can be not alleviated by using adaptive step size...". This is when the authors are talking about the adaptive learning methods. Is this a too strong comment about the adaptive learning methods? Can the authors know for sure that these methods cannot be used here?
3) Equations (1) and (2): Are they correct? should the minus sign just be in front of the first term \mathcal{M}_t(u) only?
4) Page 4, first line after Def 3: "one covers the usual gradient \nabla\rho_\theta(x)". It is not very clear (to me) how to get this. Can the authors please elaborate more on this?





**Experience Assessment:**

I have read many papers in this area.

**Review Assessment: Checking Correctness Of Derivations And Theory:**

I assessed the sensibility of the derivations and theory.

**Review Assessment: Checking Correctness Of Experiments:**

I assessed the sensibility of the experiments.

**Review Assessment: Thoroughness In Paper Reading:**

I read the paper thoroughly.

---

> ### Author Response · Authors · 2019-11-15
> **Reply**
>
>  Thanks for your comments.
> It is indeed the case that the considered estimation problem is hard in general, especially as the dimension of the model increases. However, when the model enjoys some regularity properties, the statistical estimation becomes easier. The proposed estimator allows to exploit the regularity of model, whenever possible, to provide a relatively cheap and yet accurate approximation of the natural gradient as shown in the well-specified case (Thm 14) and ill-specified case (Thm 7).
> In the paper, the final estimator results from several approximation steps, each one of those steps results in an additional error that needs to be quantified, this is precisely what the two theorems achieve.
>
>
> 1- Choice of  the kernel: It  defines the rkhs and thus depends on the smoothness of the problem. For the problems considered, we found that the rational quadratic kernel worked as well as the gaussian kernel (Figure 7 ).
>
> 2- Choice of the bandwidth: we propose a heuristic for choosing the bandwidth in section 3.3 which is of the order of the mean distance between samples and basis points.  We found that it worked well in practice and essentially avoided saturation effects due to exponentiation.  In addition we included a figure 5 shows that the estimator is accurate in a wide range of bandwidths for the 3 synthetic data considered:(normal, log-normal, and uniform on a hyper-sphere). The range where the estimator is accurate depends however on the chosen model: hyper-sphere needs small bandwidth while normal and log-normal requires large bandwidths.
>
> 3- Invariance property: Proposition 1 explicitly explains what it is.
>
> 4-  Adaptive learning methods: we changed the sentence to: “Using adaptive step sizes can help when the principal directions of curvature are aligned with the coordinates of the vector parameters. Otherwise, an additional rotation of the basis is needed to achieve this alignment.”
> Adaptive learning methods are indeed powerful optimizers when combined with a suitable parametrization of the model. They perform a diagonal scaling of the gradient to adjust the step-size for each dimension. When the principal directions of curvature matches the coordinates of the gradient this approach would be effective, however these principal direction need not be aligned with the coordinates of the gradient in general. In this case the rotation of the gradient is also needed to first align it with the principal directions of curvature. This doesn’t prevent from using adaptive methods as they can still help, but using a non-diagonal conditioning   (rotation + scaling)  could be more effective than (scaling) especially when the curvature varies a lot from one direction to another.
>
> 5- Equation 1 and 2 are correct, the minus sign compensate for another minus sign that appears after solving the minimization problem. Having the minus sign inside would work as well.
>
> 6- Distributional gradient: when the model has  a density that is smooth, the Distributional gradient defined in Def 3. admits an expression in terms of the gradient of the density. We added  Proposition 12 in appendix C.1 to make it more rigorous. This is a consequence of the  general result on derivatives under the integral.

---

### Public Comment · ~XY_Tian1 · 2019-11-07
**The algorithm's performance is much worse than SGD, Adam, and any common stochastic optimization algorithm**

Initially, this paper was very interesting to me. I tried to implement the algorithms and apply the proposed algorithms to train other models. I cannot reproduce figures 3 and 4. I wish the others and authors can check these results carefully. Also, these results are very misleading, SGD + Momentum + WD with proper learning rate decay can get much better results than the reported ones. Moreover, such algorithms should also be tested on the ImageNet. I tried it, but the performance is poor compared with any common stochastic optimization algorithm.

---

> ### Author Response · Authors · 2019-11-07
> **Reply**
>
> Hi XY, thank you for your interest in our paper, and for your work in checking the reproducibility of our results.  We are aware that there are some subtleties in the implementation, and we’d be happy to help you get the method running. Very importantly, there is a version of the algorithm which is more stable numerically and involves first performing an svd on CC^T = US^t and then pre-multiplying T by S^{\dagger}U^tT, this leads to a simplified expression which enjoys a better conditioning. We will be giving the code of our core algorithm in a separate review reply comment. We suggest that you compare this with your code, and run a sanity check that this works as expected to reproduce the Figure 1 result. If so, then try Figures 3 and 4. We will shortly be posting the anonymised code to run the entire fig. 3 and 4 experiments to the rebuttal thread, as soon as we have it ready. Meanwhile, we’re happy to answer any further questions.
> - Regarding SGD+Momentum+WD, are the performances that you mention achieved for the well conditioned case (c,d) or for the ill-conditioned one (a,b), on which dataset? For the well conditioned case SGD+momentum achieves the best performance on cifar10 (93.5 test accuracy) which we think matches the performance usually reported for the network used here, Resnet18. The same holds for cifar100. In the ill-conditioned case, we noticed a drop in performance as shown in (a,b), we think it is due to the WD and expect that a smaller weight decay would improve the performance as  it is the case for SGD+Momentum (green plot, 72% test accuracy) where the weight decay is set to 0. Is this consistent with the results you have? Could you please tell us what settings you used, and we’d be happy to investigate further.
> - Thank you for suggesting Imagenet as an additional dataset. There is an important consideration to take into account which is the dimensionality of the logits. In the case of Imagenet it is 1000 which makes it much higher dimensional compared to cifar10 or cifar100. We think that the proposed estimator would require many more samples to get accurate estimations in the case of imagenet. The theory predicts that the number of basis points M should scale linearly with the output dimension of the network (Theorem 5). While, Theorem 5 is an asymptotic result in (5), a finite sample size bound in N (Proposition 12) also shows that N is required to be greater than a certain threshold that grows linearly with the output dimension. This is also confirmed in the experiments:  In figure 1, the error increases as the dimension of the model increases. In Figure 3 and 4, there is  a drop in performance as we move from cifar10 to cifar100 dataset. On the other hand, the performance on cifar100 increases when the batch-size is increased from 128 to 512.
> - More generally, if the model has already a good conditioning one should expect little benefit in using second order methods which are generally more sensitive and require additional parameters like the damping.

---

> > ### Public Comment · ~XY_Tian1 · 2019-11-08
> > **Thank you for your reply! But I believe my previous judgement is correct!**
> >
> > Thank you for clarifying your algorithm. I implemented exact the same thing. But the performance is very bad. As the authors mentioned about the ill-conditioned cases? Why did not compare with ADAM and RMSPROP? SGD with appropriate hyperparameter can even solve such problem.
> >
> > After testing the proposed Wasserstein Natural Gradient (NGD), I found it is very problematic. The reported results almost did not compare with any popular methods.

---

> > > ### Author Response · Authors · 2019-11-08
> > > **Code to reproduce the results**
> > >
> > > We're sorry to hear you had issues implementing the results. We have posted the code to obtain the figures at the anonymised link: https://www.dropbox.com/sh/2tquwlji582lk5x/AAB8L6p8ZnBF67CscKTIg3kea?dl=0.
> > > You should be able to obtain the results of our method with this code. if you still have difficulties obtaining the figures with our code, then please share your code with us on the openreview site, and we'll do our best to help.
> > > Since submission, we have improved our results with better parameter choice, so the results will be improved over those posted in the figures.

---

### Author Response · Authors · 2019-11-15
**New revision**

We thank all reviewers for the careful review and helpful feedback. We have implemented the suggested clarifications in the revised version (details below),  and have run the following additional experiments:

1- Sensitivity analysis to the bandwidth of the kernel on the synthetic data in Figure 5 of appendix D.2..
2- Comparison between two different choices of the kernel on cifar10 (gaussian vs rational quadratic) in Figure 7 of appendix D.1
3- Comparison with Adam on Cifar10 in Figure 3 of the main text.
4- Timing comparison on Cifar10 between the compared methods in Figure 6 of appendix D.1.
5- Sensitivity to the choice of the diagonal regularization term D and a comparison with a baseline using only a diagonal preconditioning with matrix D all in Figure 7 of appendix D.1.
6- An additional synthetic dataset consisting of hyper-spheres parametrized by their center and radius in Figure 1 of the main text. This model doesn’t admit a density and is therefore more illustrative for the cases when the Wasserstein natural gradient can be used while the fisher is not well-defined.

In addition to that we made the following changes to the revised version:

1- A high level introduction of  the natural gradient using a perturbative approach in section 2.1.
2- A result showing the invariance to parameterization of natural gradient in the continuous-time limit in Proposition 1.
3- A short discussion with the connection with the Negative Sobolev distance and the work of Mroueh 2019 at the end of section 2.2 + a more detailed discussion in Appendix B.3
4-  A discussion in section 3.3 about a more stable version of the proposed estimator, which we used in the experiments and which holds when lambda = 0 (ridgeless regression).
5- A modification of proposition 4 to cover the case where lambda =0
6- A short discussion in section 3.3 about a heuristic for the choice of  the bandwidth of the kernel.
7- We also deferred the ‘well specified’ case in section 3.4 to the appendix due to space constraints.

Finally, we now better highlight the fact that we have implemented a comparison with the Fisher natural gradient on Cifar10 and Cifar100, using the KFAC and eKFAC optimizers, which compute an approximation to the Fisher gradient. However, we did not yet implement the estimator of the Fisher gradient suggested by the variational expression of proposition 2, which is an interesting topic for future work.

We would also like to bring to the attention of the reviewers that the proposed algorithm is competitive in terms of computational cost and overall time as shown in Figure 6 of Appendix D.1.

---

### Public Comment · ~XY_Tian1 · 2019-12-21
**Does the proposed method has any advantage? REAL OR FAKE RESULTS?**

The method is shown to outperform various other optimizers on a neural net optimization problem that's artificially made ill-conditioned?

REALLY? SGD with Momentum or ADAM perform much better than the proposed method. I do not know if the presented results are real or FAKE.

I could not believe how this paper was reviewed!

---

> ### Comment · Program_Chairs · 2020-11-17
> **This is not a constructive comment**
>
> Much more useful would be to discuss with the authors here. Attacking others is generally not a good practice. Instead why not provide some constructive criticism or start a discussion? Further, when giving criticism, there should be support for your claims. I hope you edit your comment.

---

### Decision · Program_Chairs · 2019-12-19

**Decision:**

Accept (Spotlight)

**Comment:**

This is a very interesting paper which extends natural gradient to output space metrics other than the Fisher-Rao metric (which is motivated by approximating KL divergence). It includes substantial mathematical and algorithmic insight. The method is shown to outperform various other optimizers on a neural net optimization problem that's artificially made ill-conditioned; while it's not clear how practically meaningful this setting is, it seems like a good way to study optimization. I think this paper will be of interest to a lot of researchers and could open up new research directions, so I recommend acceptance as an Oral.